# Partial-Label Learning with a Reject Option

**Tobias Fuchs**  *tobias.fuchs@kit.edu*
*Karlsruhe Institute of Technology, Germany*

**Florian Kalinke**  *florian.kalinke@kit.edu*
*Karlsruhe Institute of Technology, Germany*

**Klemens Böhm**  *klemens.boehm@kit.edu*
*Karlsruhe Institute of Technology, Germany*

**Reviewed on OpenReview:** *https://openreview.net/forum?id=wS1fDOofay*

## Abstract

In real-world applications, one often encounters ambiguously labeled data, where different annotators assign conflicting class labels. Partial-label learning allows training classifiers in this weakly supervised setting, where state-of-the-art methods already show good predictive performance. However, even the best algorithms give incorrect predictions, which can have severe consequences when they impact actions or decisions. We propose a novel risk-consistent nearest-neighbor-based partial-label learning algorithm with a reject option, that is, the algorithm can reject unsure predictions. Extensive experiments on artificial and real-world datasets show that our method provides the best trade-off between the number and accuracy of non-rejected predictions when compared to our competitors, which use confidence thresholds for rejecting unsure predictions. When evaluated without the reject option, our nearest-neighbor-based approach also achieves competitive prediction performance.

## 1 Introduction

Real-world data is often noisy, for example, human annotators might assign different class labels to the same instance. In partial-label learning (PLL; Hüllermeier & Beringer 2005; Liu & Dietterich 2012; Zhang & Yu 2015; Xu et al. 2023), training instances can have multiple labels, known as candidates, of which only one is correct. While in some cases it is possible to sanitize such data, cleaning is costly, especially for large-scale datasets. Instead, one wants to predict the class labels of unseen instances having sets of candidates only, that is, without knowing the exact ground-truth labels of the training data. PLL algorithms allow for handling such ambiguously labeled data.

However, even the best algorithms give incorrect predictions. These errors can have severe consequences when they impact actions or decisions. Consider, for example, safety-critical domains such as the classification of medical images (Yang et al., 2009; Lambrou et al., 2011; Senge et al., 2014; Kendall & Gal, 2017; Reamaroon et al., 2019) or the control of self-driving cars (Xu et al., 2014; Varshney & Alemzadeh, 2017; Hubmann et al., 2017; Shafaei et al., 2018; Michelmore et al., 2020). One option to limit fallacies is to employ so-called *reject options*, which allow one to abstain from certain predictions if unsure and, instead, let humans decide on the label of an instance or the actions to take (Mozannar et al., 2023). Naturally, there is a trade-off arising between the number and accuracy of non-rejected predictions. In the supervised setting, reject options have already been studied, both, for multi-class classification (Charoenphakdee et al., 2021; Cao et al., 2022; Mao et al., 2024; Narasimhan et al., 2024) and regression tasks (Zaoui et al., 2020; Cheng et al., 2023).

In the weakly supervised PLL setting, obtaining sensible reject options is more challenging than in the supervised case as ground truth is not available. Still, a reject option allows for mitigating misclassifications

and improving prediction quality. Determining whether to reject a prediction of a PLL algorithm is difficult as the uncertainty involved in prediction-making originates from multiple sources (Hora, 1996; Sale et al., 2023; Wimmer et al., 2023). On the one hand, there is inherent uncertainty due to ambiguously labeled data (*aleatoric uncertainty*). This ambiguity is influenced by both instances and classes: In some regions of the instance space, annotators are more likely to mislabel instances than in others. Additionally, some class labels are inherently more similar than others. On the other hand, there is uncertainty due to the lack of knowledge regarding the relevance of each candidate label (*epistemic uncertainty*). To the best of our knowledge, we are the first to study reject options in the context of PLL.

We propose a novel partial-label learning approach based on Dempster-Shafer theory (DST; Dempster 1967; Shafer 1986). In contrast to existing PLL approaches, which use point estimates of the candidate label weights (Hüllermeier & Beringer, 2005; Liu & Dietterich, 2012; Zhang & Yu, 2015; Ni et al., 2021; Xu et al., 2023), our method maintains a feasible region, also known as *credal set*. Maintaining such a credal set is beneficial in assessing the certainty of predictions as our experiments show.

**Contributions.** We summarize our contributions as follows.

- *Algorithm with reject option.* We introduce DST-PLL, a novel nearest-neighbor-based partial-label learning algorithm with a reject option that learns from ambiguously labeled data. The algorithm effectively determines whether to reject a prediction and provides the best trade-off between the number and accuracy of non-rejected predictions when compared to the state-of-the-art PLL methods.

- *Experiments.* Extensive experiments on artificial and real-world data support our claims. We make our code and data openly available.[1]

- *Theoretical analysis.* We analyze DST-PLL, give a closed-form expression of its expected decision boundary under mild assumptions, and prove its risk consistency. The runtime analysis shows that the proposed method's runtime is dominated by $k$-nearest-neighbor search, which has an average time complexity of $\mathcal{O}(dk \log n)$, with $d$ features and $n$ training instances.

**Structure of the paper.** We discuss related work in Section 2, define the problem setting in Section 3, and propose our method in Section 4. Section 5 features experiments and Section 6 concludes. All proofs and additional experiments are in the appendices.

## 2 Related Work

Imperfect data often renders the application of supervised methods challenging. Weakly supervised learning tackles this setting and encompasses a variety of problem formulations (Bylander, 1994; Hady & Schwenker, 2013; Ishida et al., 2019) including PLL. We discuss related work regarding PLL in Section 2.1. Section 2.2 elaborates on related work in dealing with uncertainty in ML through the lens of reject options.

### 2.1 Partial-Label Learning

The early PLL approaches transfer common supervised learning frameworks to the PLL context: Grandvalet (2002) proposes a logistic regression formulation, Jin & Ghahramani (2002) propose an expectation-maximization strategy, Hüllermeier & Beringer (2005) propose a $k$-nearest-neighbors method, Nguyen & Caruana (2008) propose an extension of support-vector machines, and Cour et al. (2011) introduce an average loss formulation allowing for the use of any supervised method. However, these approaches (i) cannot model the relevance of candidate labels in the labeling process or (ii) are not robust to non-uniform noise in the candidate sets. We emphasize that (i) and (ii) stem from two different sources of uncertainty: (i) arises from the lack of knowledge about the candidate label relevancies of the PLL model (epistemic uncertainty) and (ii) arises from the random label noise in the candidate sets and is inherent to the PLL problem (aleatoric uncertainty).

---

[1] https://github.com/mathefuchs/pll-with-a-reject-option.

More recent approaches address (i) and (ii) in the following ways: Zhang & Yu (2015); Zhang et al. (2016); Xu et al. (2019); Wang et al. (2019); Feng & An (2019); Ni et al. (2021) leverage ideas from representation learning (Bengio et al., 2013), Yu & Zhang (2017); Wang et al. (2019); Feng & An (2019); Ni et al. (2021) extend the maximum-margin idea, Liu & Dietterich (2012); Lv et al. (2020) propose extensions of the expectation-maximization strategy, Zhang et al. (2017); Tang & Zhang (2017); Wu & Zhang (2018) propose stacking and boosting ensembles for more robust prediction-making, Lv et al. (2020); Cabannes et al. (2020) introduce a minimum loss formulation to iteratively remove noise, and Feng et al. (2020); Lv et al. (2020); Xu et al. (2021); Wang et al. (2022); He et al. (2022); Xu et al. (2023); Fan et al. (2024) employ deep-learning techniques such as auto-encoders, data augmentation, and transformers in the context of PLL. These methods address (i) and (ii), featuring already good performance. However, considering a reject option of those methods based on a confidence threshold, as proposed in the supervised setting (Ni et al., 2019), yields sub-optimal results as all of the listed methods mix uncertainty stemming from (i) and (ii). We claim that keeping both separate provides a better trade-off for the number and accuracy of non-rejected predictions (Section 5).

In the PLL context, we consider (Hüllermeier & Beringer, 2005; Zhang & Yu, 2015; Zhang et al., 2016; Xu et al., 2019; Wang et al., 2019) as most closely related to our proposed method as these approaches consider an instance's neighborhood to infer its class label. To the best of our knowledge, we are the first to propose an extension of the $k$-NN classifier that keeps uncertainty from (i) and (ii) separate by leveraging Dempster-Shafer theory.

## 2.2 Reject Options

Recently, much attention has been given to the study of reject options (Mozannar et al., 2023; Mao et al., 2024; Narasimhan et al., 2024). A reject option allows one to abstain from predictions and defer them to humans rather than making uncertain and possibly harmful decisions.

There are two common strategies in rejecting predictions in the supervised setting: The confidence-based and the classifier-rejector approach (Ni et al., 2019; Cao et al., 2022). The confidence-based strategy uses a threshold on the models' confidence in order to accept or reject predictions. Common model choices for quantifying the confidences are Bayesian methods (Kingma & Welling, 2014; Kendall & Gal, 2017) and ensembles (Lakshminarayanan et al., 2017; Wimmer et al., 2023). In contrast, the classifier-rejector approach jointly learns the classifier and rejector (Ni et al., 2019; Mao et al., 2024), which can have beneficial theoretical properties. However, the classifier-rejector approach is less flexible than the confidence-based strategy as it is coupled with the concrete loss formulation of the classifier. We propose an extension of the confidence-based strategy for partial-label learning in Section 4.2.

Calibration methods (Naeini et al., 2015; Guo et al., 2017; Ao et al., 2023) are also related to the confidence-based rejection strategy as both are used to make statements about the certainty of predictions. While reject options provide a binary decision, calibration methods modify the predicted confidences such that they align with the observed accuracies. In this sense, both approaches are orthogonal and cannot directly be compared. In our work, we focus on reject options.

## 3 Problem Statement and Notations

This section formally defines the partial-label learning problem with reject option, establishes notations used throughout this work, and summarizes Dempster-Shafer theory.

## 3.1 Partial-Label Learning (PLL) with Reject Option

Let $\mathcal{X} = \mathbb{R}^d$ denote a $d$-dimensional real-valued feature space and $\mathcal{Y} = [l] := \{1, \ldots, l\}$ the finite set of $3 \leq l \in \mathbb{N}$ classes. A partial-label learning training dataset $\mathcal{D} = \{(x_i, s_i) \mid i \in [n]\}$ of $n$ instances contains feature vectors $x_i \in \mathcal{X}$ and a set of candidate labels $s_i \subseteq \mathcal{Y}$ for each $i \in [n]$. All instances $i$ have an unknown ground-truth label $y_i \in \mathcal{Y}$, and $y_i \in s_i$. Further, the candidate labels $s_i$ can be partitioned into $s_i = \{y_i\} \cup z_i$ with $y_i \notin z_i$, that is, $z_i \subseteq \mathcal{Y} \setminus \{y_i\}$ are the false-positive labels.

The sample space is $\Omega = \mathcal{X} \times \mathcal{Y} \times 2^{\mathcal{Y}}$. It is equipped with the Borel $\sigma$-algebra $\mathcal{B}(\Omega)$, yielding the measurable space $(\Omega, \mathcal{B}(\Omega))$ with respect to which the set of all probability measures $\mathcal{M}_1^+(\Omega, \mathcal{B}(\Omega))$ is defined. Let $(X, Y, S) \sim \mathbb{P} \in \mathcal{M}_1^+(\Omega, \mathcal{B}(\Omega))$, where the random variables $X : \Omega \to \mathcal{X}$, $Y : \Omega \to \mathcal{Y}$, and $S : \Omega \to 2^{\mathcal{Y}}$ govern an instance's features, true label, and candidate labels, respectively. We denote the corresponding marginal measures by associating the corresponding index to $\mathbb{P}$, for example, $\mathbb{P}_X$ is the distribution of $X$. $\mathbb{P}_{XY}$ denotes the joint distribution of $(X, Y)$.

PLL aims to train a classifier $g : \mathcal{X} \to \mathcal{Y}$ that, given a measurable loss function $\mathcal{L} : \mathcal{Y} \times 2^{\mathcal{Y}} \to \mathbb{R}_{\geq 0}$, minimizes the risk $\mathcal{R}(g) = \mathbb{E}_{(X,S) \sim \mathbb{P}_{XS}} \mathcal{L}(g(X), S)$. Common choices for $\mathcal{L}$ include the average loss (Nguyen & Caruana, 2008; Cour et al., 2011) and the minimum loss (Lv et al., 2020; Xu et al., 2021). The empirical version of the risk substitutes the expectation with a sample mean, that is, $\hat{\mathcal{R}}(g) = \frac{1}{n} \sum_{i=1}^{n} \mathcal{L}(g(x_i), s_i)$.

As misclassifications can be quite harmful, we look at the possibility of rejecting predictions, that is, abstaining from making these predictions and, instead, deferring the decisions to humans. In the PLL setting, a reject option $\Gamma_g : \mathcal{X} \to 2^{\mathcal{Y}}$ associated with a trained classifier $g$ either returns $g$'s prediction (*accept*) or abstains from making any prediction at all (*reject*), that is, $\Gamma_g(x) \in \{\emptyset, \{g(x)\}\}$ for $x \in \mathcal{X}$, with $\Gamma_g(x) = \emptyset$ denoting a reject. We then define the rejection probability $\mathrm{r}(\Gamma_g) = \mathbb{P}_X(\Gamma_g(X) = \emptyset)$ and the expected error $\mathrm{err}(\Gamma_g) = \mathbb{E}_{(X,S) \sim \mathbb{P}_{XS}}[\mathcal{L}(g(X), S)\mathbb{1}_{\{\Gamma_g(X) \neq \emptyset\}}]$ of accepted predictions. Naturally, a trade-off arises between the number and accuracy of accepted predictions, which leads us to the risk

$$\mathcal{R}_\lambda(\Gamma_g) = \mathbb{E}_{(X,S) \sim \mathbb{P}_{XS}}[\mathcal{L}(g(X), S)\mathbb{1}_{\{\Gamma_g(X) \neq \emptyset\}} + \lambda \mathbb{1}_{\{\Gamma_g(X) = \emptyset\}}] = \mathrm{err}(\Gamma_g) + \lambda \, \mathrm{r}(\Gamma_g), \tag{1}$$

for $\lambda \geq 0$; $\mathcal{L}$ characterizes the cost of misclassification and parameter $\lambda$ the cost of rejecting a prediction. In order to make well-informed decisions about when to reject, we are using Dempster-Shafer theory, which we elaborate on in the following.

## 3.2 Dempster-Shafer Theory (DST)

Dempster-Shafer theory (DST; Dempster 1967; Shafer 1986) allows for dealing with uncertainty by assigning probability mass to sets of events without specifying the probabilities of individual labels; incorrect labels do not obtain any probability mass. DST builds upon two core quantities, so-called *belief* and *plausibility*. Informally, belief collects all evidence that supports a hypothesis and plausibility collects all evidence that does not contradict a hypothesis. We argue that DST is a perfect fit for partial-label learning as one may interpret the ambiguous candidate sets as evidence regarding a label hypothesis. We further exploit belief and plausibility to inform our reject option, taking into account the difference between the supporting and non-conflicting evidence. In contrast, existing PLL approaches (Hüllermeier & Beringer, 2005; Cour et al., 2011; Liu & Dietterich, 2012; Zhang & Yu, 2015; Ni et al., 2021; Xu et al., 2023) initially assign some probability mass to each label candidate and subsequently refine them. By doing so, most probability mass is first allocated to labels that are certainly incorrect, as only one candidate is the true label. This known method renders handling the noise coming from incorrect candidate labels challenging.

With these intuitions, we now recall DST formally. In DST, a basic probability assignment (bpa) $\mathrm{m} : 2^{\mathcal{Y}} \to [0, 1]$ assigns probability mass to subsets of $\mathcal{Y}$. $\mathrm{m}$ satisfies $\mathrm{m}(\emptyset) = 0$ and $\sum_{A \subseteq \mathcal{Y}} \mathrm{m}(A) = 1$. This differs from standard probability as $\mathbb{P}(\mathcal{Y}) = 1$ for any $\mathbb{P} \in \mathcal{M}_1^+(\mathcal{Y}, 2^{\mathcal{Y}})$ but $\mathrm{m}(\mathcal{Y}) \leq 1$. Also, the mass allocated to non-intersecting sets does not necessarily add up to the mass allocated to the union, that is, one can have $\mathrm{m}(\{1, 2\}) \neq \mathrm{m}(\{1\}) + \mathrm{m}(\{2\})$. In this sense, DST allows for more flexibility as one can allocate mass on the set $\{1, 2\}$ without needing to specify any mass for $\{1\}$ and $\{2\}$ if uncertain. The sets $A \subseteq \mathcal{Y}$ with $\mathrm{m}(A) > 0$ are called *focal sets* of $\mathrm{m}$. The mass allocated to the set of all possible alternatives $\mathrm{m}(\mathcal{Y})$ can be interpreted as the degree of ignorance; it is the mass not supporting a specific alternative within $\mathcal{Y}$.

The basic probability assignment $\mathrm{m}$ does not induce a single probability measure $\mathbb{P}$ on $(\mathcal{Y}, 2^{\mathcal{Y}})$ but rather a set of probability measures $\mathcal{C}_{\mathrm{m}}(\mathcal{Y}, 2^{\mathcal{Y}})$, which is called *credal set* (Abellán et al., 2006; Cuzzolin, 2021). The probability measures $\mathbb{P} \in \mathcal{C}_{\mathrm{m}}(\mathcal{Y}, 2^{\mathcal{Y}})$ are restricted by imposing lower and upper bounds, which are called *belief* and *plausibility*, respectively. They are defined as follows:

$$\mathrm{bel}_{\mathrm{m}}(A) := \sum_{B \subseteq A} \mathrm{m}(B) \quad \text{and} \quad \mathrm{pl}_{\mathrm{m}}(A) := 1 - \mathrm{bel}_{\mathrm{m}}(\mathcal{Y} \setminus A) = \sum_{B \subseteq \mathcal{Y}, A \cap B \neq \emptyset} \mathrm{m}(B) \quad \text{for } A \in 2^{\mathcal{Y}}. \tag{2}$$

Recall that *belief* collects all evidence that supports a hypothesis $A \in 2^{\mathcal{Y}}$ (or a more specific one $B \subseteq A$), and *plausibility* collects all evidence that does not contradict a hypothesis $A \in 2^{\mathcal{Y}}$ (or a more specific one $B \subseteq A$). With belief and plausibility as above, the set of all probability measures supporting m is

$$\mathcal{C}_{\mathrm{m}}(\mathcal{Y}, 2^{\mathcal{Y}}) := \left\{ \mathbb{P} \in \mathcal{M}_1^+(\mathcal{Y}, 2^{\mathcal{Y}}) \mid \mathrm{bel}_{\mathrm{m}}(A) \le \mathbb{P}(A) \le \mathrm{pl}_{\mathrm{m}}(A) \text{ for all } A \subseteq \mathcal{Y} \right\} \subseteq \mathcal{M}_1^+(\mathcal{Y}, 2^{\mathcal{Y}}). \tag{3}$$

Further, DST provides rules to combine m-s from multiple sources (Dempster, 1967; Yager, 1987a;b). This is beneficial in the PLL setting as there is several conflicting evidence about the class labels within a neighborhood of instances. Estimating a credal set $\mathcal{C}_{\mathrm{m}}(\mathcal{Y}, 2^{\mathcal{Y}})$ from such a neighborhood allows us to construct an effective reject option, which we detail in Section 4.2.

Several methods already leverage DST in supervised learning (Mandler & Schümann, 1988; Denoeux, 1995; Tabassian et al., 2012; Sensoy et al., 2018; Denoeux, 2019; Tong et al., 2021). We consider the nearest neighbor approach by Denoeux (1995) to be most closely related to our approach. Here, basic probability assignments are constructed from the nearest neighbors of an instance. Then, Dempster's rule is used to combine them into a single bpa. Their analysis is, however, not transferable to our case because they only have singletons or the full label space as focal sets, making set intersections in the combination rule easy to handle. In this sense, we examine a more general setting since we allocate probability mass to arbitrary subsets.

## 4 Our Method: DST-PLL

This section introduces our novel partial-label learning method DST-PLL. Based on the labeling information of an instance's nearest neighbors, we construct basic probability assignments within Dempster-Shafer theory. These bpas inform the prediction and rejection decisions as discussed in Section 4.1 and Section 4.2, respectively. Regarding the reject option, we propose a novel variation of the confidence-based rejection strategy: The confidence threshold is adaptively selected on a per-instance basis dependent on the amount of incorrect label noise. The more noise from incorrect labels there is, the more confident the model needs to be to accept a prediction.

Algorithm 1 outlines DST-PLL, which we summarize in the following. We denote by $\mathrm{NN}_k(\tilde{x}) \subseteq \mathcal{X} \times 2^{\mathcal{Y}}$ the set of the $k$-nearest neighbors of instance $\tilde{x}$ with their associated candidate labels. To predict the class label of an instance $\tilde{x}$ (Line 11), the algorithm first transforms information from $\tilde{x}$'s neighbors $\mathrm{NN}_k(\tilde{x})$ into bpas $\mathrm{m}_i$ (Lines 3–7), collects these into evidence set $\mathcal{E}$ (Line 8), and combines the bpas into $\tilde{\mathrm{m}}$ using Yager's rule (Line 10; Yager 1987a;b). Section 4.1 elaborates on these steps. Section 4.2 elaborates on how we extract our reject option from $\tilde{\mathrm{m}}$ (Line 12). Section 4.3 shows that the resulting classification rule is risk-consistent. We analyze our algorithm's runtime in Section 4.4.

### 4.1 Making Predictions

**Basic probability assignments.** Following the standard assumption that neighboring instances in feature space are also close in label space, we combine the evidence from the $k$-nearest neighbors $(x_i, s_i) \in \mathrm{NN}_k(\tilde{x})$ of a given instance $\tilde{x} \in \mathcal{X}$ with its candidate labels $\tilde{s} \subseteq \mathcal{Y}$ ($\tilde{s} = \mathcal{Y}$ if $\tilde{x}$ is a test instance).

When looking at a neighboring instance $(x_i, s_i) \in \mathrm{NN}_k(\tilde{x})$, there are generally two cases: either (i) $(x_i, s_i)$ provides information about the correct label of $\tilde{x}$ or (ii) $(x_i, s_i)$ is irrelevant for finding the correct label of $\tilde{x}$. To address (i), we allocate probability mass on the candidates $s_i$ of neighbor $x_i$. To address (ii), we allocate probability mass on the full label space $\mathcal{Y}$ indicating uncertainty about the correct label of $\tilde{x}$.

More formally, for fixed $i \in [k]$, the candidate labels $s_i$ do not provide any valuable information if they support all ($\tilde{s} \subseteq s_i$) or none ($\tilde{s} \cap s_i = \emptyset$) of the labels in $\tilde{s}$; we use a bpa of $\mathrm{m}_i(\tilde{s}) = 1$ (Line 4). We set $\mathrm{m}_i(A) = 1/2$ if $A = \tilde{s}$ or $A = \tilde{s} \cap s_i$, else $\mathrm{m}_i(A) = 0$ (Line 6), where $1/2$ equally weights evidence. We later elaborate further on this choice and demonstrate the application of the proposed classification rule in Example 4.1. Note that we make the common assumption that the true label of instance $\tilde{x}$ is always in $\tilde{s}$ (Cour et al., 2011; Liu & Dietterich, 2012; Lv et al., 2020; Ni et al., 2021). While our definition of the $\mathrm{m}_i$-s is similar to (Denoeux,

---

**Algorithm 1** DST-PLL (Our proposed method)

---

**Input:** Partially-labeled dataset $\mathcal{D} = \{(x_i, s_i) \in \mathcal{X} \times 2^{\mathcal{Y}} \mid i \in [n]\}$, number of nearest neighbors $k$, instance $\tilde{x} \in \mathcal{X}$ for inference with candidate labels $\tilde{s} \subseteq \mathcal{Y}$ ($\tilde{s} = \mathcal{Y}$ if $\tilde{x}$ is an unseen test instance);

**Output:** Prediction $g(\tilde{x})$ and reject option $\Gamma_g(\tilde{x})$ for instance $\tilde{x}$;

1: $\mathcal{E} \leftarrow \emptyset$
2: **for** $(x_i, s_i) \in \mathrm{NN}_k(\tilde{x})$ **do**
3:    **if** $\tilde{s} \subseteq s_i$ or $\tilde{s} \cap s_i = \emptyset$ **then**
4:       $\mathrm{m}_i : 2^{\mathcal{Y}} \to [0,1], \ A \mapsto \begin{cases} 1 & \text{if } A = \tilde{s}, \\ 0 & \text{else} \end{cases}$
5:    **else**
6:       $\mathrm{m}_i : 2^{\mathcal{Y}} \to [0,1], \ A \mapsto \begin{cases} 1/2 & \text{if } A = \tilde{s} \text{ or } A = \tilde{s} \cap s_i, \\ 0 & \text{else} \end{cases}$
7:    **end if**
8:    $\mathcal{E} \leftarrow \mathcal{E} \cup \{\mathrm{m}_i\}$
9: **end for**
10: $\tilde{\mathrm{m}} \leftarrow \mathrm{yager\_combination}(\tilde{s}, \mathcal{E})$
11: $g(\tilde{x}) \leftarrow \begin{cases} \arg\max_{y \in \tilde{s}} \tilde{\mathrm{m}}(\{y\}) & \text{if } \max_{y \in \tilde{s}} \tilde{\mathrm{m}}(\{y\}) > 0, \\ \text{Randomly pick from } \arg\max_{A \subseteq \tilde{s}} \tilde{\mathrm{m}}(A) & \text{else;} \end{cases}$
12: $\Gamma_g(\tilde{x}) \leftarrow \begin{cases} \{g(\tilde{x})\} & \text{if } \Delta_{\tilde{\mathrm{m}}} > 0 \text{ as defined in (5)}, \\ \emptyset & \text{else;} \end{cases}$
13: **return** $(g(\tilde{x}), \Gamma_g(\tilde{x}))$

---

1995), we target a more general setting as our focal sets can be arbitrary subsets instead of only singletons or the full label set.

The bpa $\mathrm{m}_i$ has the following four effects on belief and plausibility as defined in (2): (i) A set of candidates $A$ has maximal belief, that is, $\mathrm{bel}_{\mathrm{m}_i}(A) = 1$, if it covers $\tilde{s}$, that is, $\tilde{s} \subseteq A$. (ii) A set of candidates $A$ is plausible, that is, $\mathrm{pl}_{\mathrm{m}_i}(A) > 0$, if it supports at least one of the candidate labels in $\tilde{s}$, that is, $A \cap \tilde{s} \neq \emptyset$. (iii) There is a gap, that is, $\mathrm{bel}_{\mathrm{m}_i}(A) < \mathrm{pl}_{\mathrm{m}_i}(A)$, if $A$ supports some candidate in $\tilde{s} \cap s_i$ but does not cover all candidates in $\tilde{s} \cap s_i$ or supports some candidate in $\tilde{s}$ but does not cover all candidates of $\tilde{s}$. (iv) Class labels $y \in \tilde{s} \cap s_i$ are maximally plausible, that is, $\mathrm{pl}_{\mathrm{m}_i}(\{y\}) = 1$.

**Evidence weighting.** Our definition of the $\mathrm{m}_i$-s (Algorithm 1, Line 6) also permits a more general view, that is, $\mathrm{m}_i(A) = \alpha$ if $A = \tilde{s}$, $\mathrm{m}_i(A) = 1 - \alpha$ if $A = \tilde{s} \cap s_i$, and $\mathrm{m}_i(A) = 0$ otherwise, for some $\alpha \in (0, 1)$. However, without further assumptions, one cannot know how relevant the information from a particular neighbor is. The setting of $\alpha = 1/2$, which we use, weights supporting and conflicting evidence of all neighbors equally. In other words, if a neighbor's evidence excludes some candidate labels from consideration, it is of equal importance compared to supporting some candidate labels. Therefore, we set $\alpha = 1/2$.

**Evidence combination.** Given the set $\mathcal{E} = \{\mathrm{m}_i \mid i \in [k]\}$, we combine all $\mathrm{m}_i$-s using Yager's rule (Yager, 1987a;b). Dempster's original rule (Dempster, 1967) enforces $\tilde{\mathrm{m}}(\emptyset) = 0$ by normalization, which is criticized for its unintuitive results when facing high conflict (Zadeh, 1984). Instead, Yager's rule first collects overlapping evidence in $\mathrm{q} : 2^{\mathcal{Y}} \to [0,1]$ and creates a valid bpa $\tilde{\mathrm{m}} : 2^{\mathcal{Y}} \to [0,1]$ by

$$\mathrm{q}(A) := \sum_{\substack{A_1, \ldots, A_k \subseteq \mathcal{Y} \\ \bigcap_{i=1}^{k} A_i = A}} \prod_{j=1}^{k} \mathrm{m}_j(A_j) \quad \text{and} \quad \tilde{\mathrm{m}}(A) := \begin{cases} 0 & \text{if } A = \emptyset, \\ \mathrm{q}(\mathcal{Y}) + \mathrm{q}(\emptyset) & \text{if } A = \mathcal{Y}, \\ \mathrm{q}(A) & \text{else.} \end{cases} \tag{4}$$

We implement this efficiently with hash maps storing only the focal sets.[1]

**Classification rule.** After the combination into m̃ by (4), we extract a prediction $g(\tilde{x})$ for instance $\tilde{x}$ (Line 11). We predict the class label with the highest probability mass $\arg\max_{y \in \tilde{s}} \tilde{m}(\{y\})$ if any has non-zero mass, or else randomly pick from the subset with the most mass $\arg\max_{A \subseteq \tilde{s}} \tilde{m}(A)$. When tied, we use the subset with the smallest cardinality. In the following, we present an example of our classification rule.

*Example* 4.1 (Classification rule). Let $k = 3$, $\tilde{x}$ an unseen test instance, $(x_i, s_i) \in NN_k(\tilde{x})$, $\mathcal{Y} = \{1, 2, 3\}$, $s_1 = \{1\}$, $s_2 = \{1, 2\}$, and $s_3 = \{1, 3\}$. Then, $m_1(\{1\}) = m_1(\mathcal{Y}) = 1/2$, $m_2(\{1, 2\}) = m_2(\mathcal{Y}) = 1/2$, and $m_3(\{1, 3\}) = m_3(\mathcal{Y}) = 1/2$. All other subsets receive a mass of zero. Using Yager's combination rule, we obtain $\tilde{m}(\{1\}) = 5/8$, $\tilde{m}(\{1, 2\}) = \tilde{m}(\{1, 3\}) = \tilde{m}(\mathcal{Y}) = 1/8$, and $\tilde{m}(A) = 0$ for the remaining $A \subseteq \mathcal{Y}$. Therefore, we predict label 1.

## 4.2 Proposed Reject Option

To limit the impact of misclassification, our method provides a reject option $\Gamma_g$, that is, the algorithm can abstain from individual predictions if unsure (Algorithm 1, Line 12). Our formulation builds on the confidence-based rejection strategy (Section 2.2), that is, $\Delta := \text{conf}(g) - \theta$ with $\text{conf}(g) \in [0, 1]$ being the model's confidence and $\theta \in [0, 1]$ the confidence threshold. We adapt this setting to the PLL context by changing the confidence threshold $\theta_{\tilde{m}}$ based on the amount of noise present.

Recall from (3) that the belief and plausibility regarding m̃ act as a lower and upper bound of the probability mass, respectively. The intuition of our reject option is as follows. If the lower bound (belief) on the probability mass of our predicted label exceeds the maximal upper bound (plausibility) on the probability mass regarding any other label, we can safely make the prediction.

In other words, if there is a class label different from the predicted one that is quite plausible (high $\max_{y \in \tilde{s} \setminus \{\hat{y}\}} \text{pl}_{\tilde{m}}(\{y\})$), we require a high belief mass in order to be certain about the prediction, that is, the belief must satisfy $\text{bel}_{\tilde{m}}(\{\hat{y}\}) > \max_{y \in \tilde{s} \setminus \{\hat{y}\}} \text{pl}_{\tilde{m}}(\{y\})$. If, instead, there is no other plausible candidate label, we can be sure of our prediction with less belief mass.

We formalize this intuition in the following. Let m̃ be the resulting bpa as determined by Algorithm 1 and

$$\Delta_{\tilde{m}} := \underbrace{\text{bel}_{\tilde{m}}(\{\hat{y}\})}_{(=\text{conf}(g))} - \underbrace{\max_{y \in \tilde{s} \setminus \{\hat{y}\}} \text{pl}_{\tilde{m}}(\{y\})}_{(=\theta_{\tilde{m}})} \quad \text{with} \quad \hat{y} := \arg\max_{y \in \tilde{s}} \tilde{m}(\{y\}). \tag{5}$$

In other words, we instantiate the model's confidence with the model's belief mass $\text{conf}(g) = \text{bel}_{\tilde{m}}(\{\hat{y}\})$ of the predicted instance $\hat{y}$ and the confidence threshold $\theta_{\tilde{m}}$ based on the amount of noise regarding other labels $y \neq \hat{y}$, that is, $\theta_{\tilde{m}} = \max_{y \in \tilde{s} \setminus \{\hat{y}\}} \text{pl}_{\tilde{m}}(\{y\})$. Note that the dependence on m̃ allows for setting the threshold adaptively.

(5) satisfies several desirable properties, which we collect in Theorem 4.2 and elaborate on in the following.

**Theorem 4.2.** *Let $\mathcal{Y}$ be the label space, $\tilde{x} \in \mathcal{X}$ the instance of interest, $\tilde{s} \subseteq \mathcal{Y}$ its candidate labels ($\tilde{s} = \mathcal{Y}$ if $\tilde{x}$ is a test instance), $g(\tilde{x})$ our algorithm's prediction, and m̃ the resulting probability mass as determined by Algorithm 1. Then, the following hold:*

*(i) If $g(\tilde{x})$ has been picked randomly (Algorithm 1, Line 11, second case), then $\Delta_{\tilde{m}} \leq 0$,*

*(ii) if $\Delta_{\tilde{m}} > 0$, then $\mathbb{P}(\{g(\tilde{x})\}) > \mathbb{P}(\{y\})$ for all $\mathbb{P} \in \mathcal{C}_{\tilde{m}}(\mathcal{Y}, 2^{\mathcal{Y}})$ and $y \in \tilde{s} \setminus \{g(\tilde{x})\}$, and*

*(iii) if $\tilde{m}(\{g(\tilde{x})\}) > 1/2$, then $\Delta_{\tilde{m}} > 0$. The converse of (iii) does not hold.*

Based on these considerations, we define the accept and reject regions of our method as follows.

1. **Accept.** When $\Delta_{\tilde{m}} > 0$, we are in the *accept* region and $\Gamma_g(\tilde{x}) = \{\hat{y}\}$: The lower-bound probability of the class label $\hat{y}$ is greater than the upper-bound probability of any other class label $y \neq \hat{y}$, that is, $\mathbb{P}(\{\hat{y}\}) > \mathbb{P}(\{y\})$ for all $\mathbb{P} \in \mathcal{C}_{\tilde{m}}(\mathcal{Y}, 2^{\mathcal{Y}})$ and $y \neq \hat{y}$.

2. **Reject.** When $\Delta_{\tilde{m}} \leq 0$, we are in the *reject* region and $\Gamma_g(\tilde{x}) = \emptyset$: The lower-bound probability of the class label $\hat{y}$ is less than or equal to the upper-bound probability of another class label $y \neq \hat{y}$. There exists $\mathbb{P} \in \mathcal{C}_{\tilde{m}}(\mathcal{Y}, 2^{\mathcal{Y}})$ and $y \neq \hat{y}$ with $\mathbb{P}(\{\hat{y}\}) \leq \mathbb{P}(\{y\})$.

We remark that Theorem 4.2 (*iii*) implies that DST-PLL rejects fewer predictions than the $k$-nearest neighbor reject option by Hellman (1970), which requires more than $1/2$ of all votes to be certain. Our method can accept predictions with less than $1/2$ of all probability mass on a single class label, while still satisfying Theorem 4.2 (*ii*), that is, if a prediction is accepted, the decision remains unchanged independent of $\mathbb{P} \in \mathcal{C}_{\tilde{m}}(\mathcal{Y}, 2^{\mathcal{Y}})$. It follows that having fewer rejections does not come at the expense of an increase in the expected error. In the following, we provide an example of the proposed reject option.

*Example* 4.3 (Reject option). Assume the setting and result of Example 4.1: $\tilde{m}(\{1\}) = 5/8$, $\tilde{m}(\{1,2\}) = \tilde{m}(\{1,3\}) = \tilde{m}(\mathcal{Y}) = 1/8$, and $\tilde{m}(A) = 0$ for the remaining $A \subseteq \mathcal{Y}$. Our prediction is $\hat{y} = \arg\max_{y \in \tilde{s}} \tilde{m}(\{y\}) = 1$ with $\tilde{s} = \mathcal{Y}$. Hence, $\Delta_{\tilde{m}} = \text{bel}_{\tilde{m}}(\{\hat{y}\}) - \max_{y \in \tilde{s} \setminus \{\hat{y}\}} \text{pl}_{\tilde{m}}(\{y\}) = \text{bel}_{\tilde{m}}(\{1\}) - \text{pl}_{\tilde{m}}(\{2\}) = 5/8 - 2/8 = 3/8$. Since $\Delta_{\tilde{m}} = 3/8 > 0$, we accept the prediction $\hat{y} = 1$.

### 4.3 Consistency

Our classification rule yields a risk-consistent classifier, which we demonstrate in the following. As is common in the literature (Cour et al., 2011; Liu & Dietterich, 2012; Feng et al., 2020; Lv et al., 2020) and required to obtain statistical guarantees in the PLL setting, we fix a label (noise) distribution (Assumption 4.4) that permits further analysis of the proposed algorithm. Appendix D experimentally verifies that Assumption 4.4 is satisfied on real-world datasets.

**Assumption 4.4.** Let $\tilde{x} \in \mathcal{X}$ be the instance of interest with hidden true label $\tilde{y} \in \mathcal{Y}$ and $l = |\mathcal{Y}| \geq 3$ classes. Its $k$ partially-labeled neighbors are $(x_i, s_i) \in \text{NN}_k(\tilde{x})$. Label $\tilde{y}_c \in \mathcal{Y} \setminus \{\tilde{y}\}$ denotes the class label that co-occurs most frequently with label $\tilde{y}$ in $\tilde{x}$'s neighborhood. We assume that the true label dominates the neighborhood, that is,

$$\mathbb{P}(S = \underbrace{s}_{\substack{= \{y\} \,\cup\, z \\ \text{Partitioning valid in cases } (ii)-(iv).}}, Y = y \mid X = x_i) = \begin{cases} 0 & \text{if } s = \mathcal{Y}, \, s = \emptyset, \, y \notin s, \, y \in z, \text{ or } \tilde{y} \in z, & (i) \\ \frac{1}{2^{l-2}-1} \, p_1 & \text{if } y = \tilde{y}, \, \tilde{y}_c \in z, \text{ and } s \neq \mathcal{Y}, & (ii) \\ \frac{1}{2^{l-2}} \, p_2 & \text{if } y = \tilde{y} \text{ and } \tilde{y}_c \notin z, & (iii) \\ \frac{1}{2^{l-1}-1} \frac{1}{|s|} \, p_3 & \text{if } y \neq \tilde{y}, \, \tilde{y} \notin z, \, y \notin z, \text{ and } s \neq \emptyset, & (iv) \end{cases}$$

with $p_1, p_2, p_3 \in (0, 1)$, $p_1 + p_2 + p_3 = 1$, and $p_1 \geq p_2 \geq p_3 > 0$.

Because $p_1 + p_2 > p_3$, Assumption 4.4 implies $\mathbb{P}_{XY}(Y = \tilde{y} \mid X = x_i) > \mathbb{P}_{XY}(Y \neq \tilde{y} \mid X = x_i)$, that is, points that are close in feature space are likely to have similar class labels. We note that this assumption is related to the ambiguity degree condition by Cour et al. (2011) as both make sure that the noise labels do not overwhelm the PLL algorithm. Assumption 4.4 enforces this as $p_1 \geq p_2 \geq p_3 > 0$. Having all cases spelled out as in Assumption 4.4 benefits the proof of Lemma 4.5.

In the following, we elaborate on the four cases $(i) - (iv)$.

- If a candidate set contains all labels ($s = \mathcal{Y}$), no labels ($s = \emptyset$), or does not allow to be partitioned into true label $y$ and false-positive labels $z$, it is not a valid candidate set (Hüllermeier & Beringer, 2005; Cour et al., 2011; Liu & Dietterich, 2012; Zhang & Yu, 2015). $(i)$ handles this pathological setting by assigning zero probability.

- In $(ii)$, $\tilde{x}$'s label $\tilde{y}$ is also the label of neighbor $x_i$ and $\tilde{y}_c$, with which $\tilde{y}$ is most commonly confused, is also part of the candidate set $s_i$. There are $2^{l-2} - 1$ sets $s$ that contain $\tilde{y}$ and $\tilde{y}_c$ and are not $\mathcal{Y}$.

- In $(iii)$, $\tilde{x}$'s label $\tilde{y}$ is also the label of neighbor $x_i$. Label $\tilde{y}_c$ is not part of the candidate set $s_i$ of neighbor $x_i$. There are $2^{l-2}$ sets $s$ that contain $\tilde{y}$ but not $\tilde{y}_c$.

- In $(iv)$, neighbor $x_i$ has different labels altogether and is irrelevant for explaining the class label of instance $\tilde{x}$. There are $2^{l-1} - 1$ sets that do not contain the label $\tilde{y}$ (excluding the empty set). Each of the $|s|$ candidates can be the neighbor's correct label.

Theorem 4.6, which establishes risk-consistency, hinges on the following intermediate results in Lemma 4.5. The fact that the $m_i$-s can have arbitrary focal sets renders the direct application of Yager's rule (4) in our

analysis challenging. The proof of Lemma 4.5 mediates this issue by providing a closed-form solution for the resulting bpa $\tilde{\mathrm{m}}$.

**Lemma 4.5.** *Assume the setting of Assumption 4.4, let $\tilde{x} \in \mathcal{X}$ be the instance of interest, $\tilde{y} \in \mathcal{Y}$ its hidden true label, $\tilde{y}_c \in \mathcal{Y} \setminus \{\tilde{y}\}$ the incorrect label with which $\tilde{y}$ is most often confused, and $\tilde{\mathrm{m}}$ the resulting probability mass as determined by Algorithm 1. Then, the following hold:*

*(i)* $\mathbb{E}_{\mathbb{P}}[\tilde{\mathrm{m}}(\{\tilde{y}\}) \mid X = \tilde{x}] > 0$, *and*

*(ii)* $\mathbb{E}_{\mathbb{P}}[\tilde{\mathrm{m}}(\{\tilde{y}\}) \mid X = \tilde{x}] > \mathbb{E}_{\mathbb{P}}[\tilde{\mathrm{m}}(\{\tilde{y}_c\}) \mid X = \tilde{x}]$.

As the probability mass of the hidden true label $\tilde{y}$ is positive in expectation by *(i)*, we can reduce our analysis to the first case of our decision rule (Algorithm 1, Line 11). *(ii)* shows how Assumption 4.4 propagates when applying Yager's rule on all $k$ neighbors' $\mathrm{m}_i$-s. The label that dominates the neighborhood obtains the highest probability mass. To our knowledge, it is the first time that such a result has been shown for arbitrary focal sets in $\tilde{\mathrm{m}}$. Denoeux (1995) analyze the special case when all focal sets are singletons or the full label space.

To put Theorem 4.6 into context, we recall the concept of risk consistency (Devroye et al., 1996). The Bayes classifier is defined by $g^* = \arg\min_{g:\mathcal{X}\to\mathcal{Y}} \mathcal{R}(g)$; it has the least overall risk. Let $g_n$ be the classifier trained by Alg. 1 with $n$ instances. Its empirical risk $\hat{\mathcal{R}}(g_n)$ can be computed by substituting the expectation with a sample mean. It is risk-consistent if $\hat{\mathcal{R}}(g_n) \to \mathcal{R}(g^*)$ for $n \to \infty$ almost surely. Theorem 4.6 establishes the risk consistency of the proposed classifier.

**Theorem 4.6.** *Assume the setting of Assumption 4.4, let $g_n : \mathcal{X} \to \mathcal{Y}$ be the classifier trained by Algorithm 1 with $n$ training instances, $g^* : \mathcal{X} \to \mathcal{Y}$ the Bayes classifier, and $\tilde{x} \in \mathcal{X}$ a fixed instance with unknown true label $\tilde{y}$. Then, the following hold:*

*(i)* $\mathbb{E}_{(x_i, s_i)_{i=1}^n \overset{i.i.d.}{\sim} \mathbb{P}_{XS}} g_n(\tilde{x}) = \tilde{y}$, *and*

*(ii)* $\lim_{n\to\infty} \big(\hat{\mathcal{R}}(g_n) - \mathcal{R}(g^*)\big) = 0$ *almost surely.*

### 4.4 Runtime Complexity

We decompose the overall runtime of our approach in Algorithm 1 into $k$-times querying one nearest neighbor and creating its bpa (Lines 2–9), the cost of Yager's rule (Line 10), as well as extracting predictions in Lines 11-12. Using the ball-tree data structure (Omohundro, 1989), querying one neighbor takes $\mathcal{O}(d \log n)$ time on average. In the worst case, query time is $\mathcal{O}(dn)$. One builds a ball-tree in $\mathcal{O}(dn \log n)$ time. We construct a bpa $\mathrm{m}_i$ by storing its focal sets within a hash map and combine all $\mathrm{m}_i$-s as defined in (4). There are at most $\min(2^k, 2^l)$ focal sets of $\tilde{\mathrm{m}}$: Each $\mathrm{m}_i$ has at most two focal sets producing $2^k$ combinations and there are at most $2^l$ possible subsets of $\mathcal{Y}$. We take the minimum as both are upper bounds. Looking up a focal set in the hash-map requires $\mathcal{O}(l)$ time as the key length is variable. Extracting a prediction and the reject option then requires $\mathcal{O}(l^2)$ time. Combining the above yields a worst-case complexity of $\mathcal{O}\big(dkn + l\big(\min(2^k, 2^l) + \max(k, l)\big)\big)$. Since $k$ and $l$ are constant, the nearest-neighbor search dominates. The average total runtime of the search is $\mathcal{O}(dk \log n)$.

## 5 Experiments

Section 5.1 lists the tested methods, Section 5.2 shows our experimental setup, and Section 5.3 collects our main findings. Additional results and a description of all hyperparameters can be found in Appendix D.

### 5.1 Algorithms for Comparison

While many PLL algorithms exist (see Section 2), we focus on state-of-the-art methods commonly used in the literature. We consider ten methods: PL-KNN (Hüllermeier & Beringer, 2005), PL-SVM (Nguyen & Caruana, 2008), IPAL (Zhang & Yu, 2015), PL-ECOC (Zhang et al., 2017), PRODEN (Lv et al., 2020), CC (Feng et al.,

2020), VALEN (Xu et al., 2021), POP (Xu et al., 2023), CROSEL (Tian et al., 2024), and DST-PLL (our proposed method). Note that we decided to not consider PICO (Wang et al., 2022) as it is only applicable to image data. In our experiments, however, we consider various image and non-image datasets as discussed in Section 5.2.

We choose the parameters of all methods as recommended by their respective authors and use the same base models for all of the neural network approaches. Appendix D.2 discusses the choices of all hyperparameters in more detail. As base models, we pick the LeNet architecture (LeCun et al., 1998) for the MNIST datasets and a $d$-300-300-300-$l$ MLP (Werbos, 1974) for all other datasets. For PL-KNN and our method, we use variational auto-encoders (Kingma & Welling, 2014) to reduce the feature space dimensionality of the MNIST datasets and compute the nearest neighbors on the hidden representations. As our competitors do not provide a reject option, we use a threshold on their model's confidences, that is, the maximum probability outputs, to evaluate the trade-off between the fraction and accuracy of non-rejected predictions.

## 5.2 Experimental Setup

**Data.** Following the default protocol (Cour et al., 2011; Zhang & Yu, 2015; Lv et al., 2020; Xu et al., 2023), we conduct several experiments using datasets for supervised learning with added artificial noise as well as experiments on real-world partially-labeled data. We repeat all experiments five times to report averages and standard deviations. For the supervised datasets, we use the *ecoli* (Horton & Nakai, 1996), *multiple-features* (Duin, 2002), *pen-digits* (Alpaydin & Alimoglu, 1998), *semeion* (Buscema & Terzi, 2008), *solar-flare* (Dodson & Hedeman, 1989), *statlog-landsat* (Srinivasan, 1993), and *theorem* datasets (Bridge et al., 2013) from the UCI repository (Bache & Lichman, 2013). These datasets contain between 336 and 10 992 instances each. Also, we use the popular *MNIST* (LeCun et al., 1999), *KMNIST* (Clanuwat et al., 2018), and *FMNIST* datasets (Xiao et al., 2018), which contain 60 000 images each similar to other datasets like CIFAR-10 and CIFAR-100 (Krizhevsky, 2009). For the partially labeled data, we use the *bird-song* (Briggs et al., 2012), *flickr* (Huiskes & Lew, 2008), *yahoo-news* (Guillaumin et al., 2010), and *msrc-v2* datasets (Liu & Dietterich, 2012). They contain between 1755 and 22 762 instances.

**Noise Generation.** We use three noise generation strategies to partially label the supervised datasets: uniform, class-dependent, and instance-dependent noise. Uniform noise (Liu & Dietterich, 2012) adds three uniform random noise labels to a fraction of 70 % of all instances. Class-dependent noise (Cour et al., 2011) randomly partitions all class labels into pairs and adds the partner label as noise to 70 % of all instances having the other label. Instance-dependent noise (Zhang et al., 2021) first trains a supervised classifier $g : \mathcal{X} \to \mathcal{M}_1^+(\mathcal{Y}, 2^{\mathcal{Y}})$. Given an instance $x$ with true label $y$, a flipping probability of $\xi_{\bar{y}}(x) := g_{\bar{y}}(x) / \max_{y' \in \mathcal{Y} \setminus \{y\}} g_{y'}(x)$ for $\bar{y} \neq y$ determines which noise labels to randomly pick.

## 5.3 Results

**Prediction Performance.** Table 1 shows the average test-set accuracies and standard deviations over all UCI datasets with class- and instance-dependent noise, MNIST datasets with class- and instance-dependent noise, and the real-world datasets. The algorithms with the highest accuracies as well as the algorithms with non-significant differences using a paired t-test with level $\alpha = 0.05$ are emphasized. Our approach (DST-PLL) performs comparably to the other methods. We note that none of the methods is best across all settings. For example, CC performs best in four out of five settings but is significantly outperformed by our approach on the real-world experiments.

**Reject Option.** To compare our reject option with the other methods, we use a threshold of $\Delta_{\tilde{m}} > 0$ for our proposed approach (Algorithm 1, Line 12), a confidence threshold of 90 % for classifiers outputting a probability distribution over the class labels, and a threshold of 50 % of all votes for PL-KNN to not reject a prediction, which is in line with the reject option by Hellman (1970).

Table 2 shows the average empirical reject-option risk and standard deviation across all experiments for varying $\lambda$. We compute $\hat{\mathcal{R}}_\lambda(\Gamma_g) = \hat{\text{er}}(\Gamma_g) + \lambda \hat{\text{r}}(\Gamma_g)$ by using ground-truth information to calculate the non-reject error $\hat{\text{er}}(\Gamma_g)$ and counting the number of rejects to calculate the reject rate $\hat{\text{r}}(\Gamma_g)$. The algorithms with the lowest risks as well as the algorithms with non-significant differences using a paired t-test with level

Table 1: Average test-set accuracies ($\pm$ std.) on the UCI, MNIST, and real-world datasets. The UCI and MNIST datasets are artificially augmented with class-dependent and instance-dependent noise. The best algorithms (highest accuracy) as well as algorithms with non-significant differences are emphasized. The significance analysis uses a paired student t-test with level $\alpha = 0.05$.

| Algorithms | UCI datasets | | MNIST-like datasets | | Real-world |
|---|---|---|---|---|---|
| | Class-dep. | Inst.-dep. | Class-dep. | Inst.-dep. | |
| PL-KNN (2005) | **81.2** ($\pm$ 13.9) | 75.8 ($\pm$ 11.1) | 92.2 ($\pm$ 4.1) | 84.8 ($\pm$ 7.2) | 53.4 ($\pm$ 10.9) |
| PL-SVM (2008) | 62.3 ($\pm$ 16.3) | 43.8 ($\pm$ 16.4) | 67.5 ($\pm$ 9.8) | 47.8 ($\pm$ 8.2) | 39.1 ($\pm$ 9.6) |
| IPAL (2015) | 79.3 ($\pm$ 17.1) | 75.3 ($\pm$ 18.3) | 92.9 ($\pm$ 4.3) | 88.5 ($\pm$ 6.6) | 58.7 ($\pm$ 9.8) |
| PL-ECOC (2017) | 63.7 ($\pm$ 13.1) | 66.5 ($\pm$ 12.5) | 64.3 ($\pm$ 14.3) | 51.7 ($\pm$ 10.7) | 46.2 ($\pm$ 10.2) |
| PRODEN (2020) | **81.6** ($\pm$ 14.0) | **78.1** ($\pm$ 13.2) | **93.9** ($\pm$ 4.4) | 88.2 ($\pm$ 6.0) | **64.2** ($\pm$ 8.2) |
| CC (2020) | 81.3 ($\pm$ 14.2) | **78.8** ($\pm$ 13.6) | **93.9** ($\pm$ 4.5) | **89.6** ($\pm$ 5.7) | 49.2 ($\pm$ 29.7) |
| VALEN (2021) | 79.7 ($\pm$ 15.3) | 75.6 ($\pm$ 12.7) | 91.8 ($\pm$ 4.2) | 83.4 ($\pm$ 8.4) | **63.6** ($\pm$ 9.7) |
| POP (2023) | **81.5** ($\pm$ 14.0) | **78.1** ($\pm$ 13.1) | **93.9** ($\pm$ 4.5) | 88.1 ($\pm$ 6.0) | **63.6** ($\pm$ 8.4) |
| CROSEL (2024) | 79.9 ($\pm$ 17.2) | **78.4** ($\pm$ 14.5) | **94.2** ($\pm$ 4.5) | **88.9** ($\pm$ 6.7) | 46.3 ($\pm$ 28.9) |
| DST-PLL (ours) | **80.8** ($\pm$ 14.1) | 75.4 ($\pm$ 10.7) | 92.0 ($\pm$ 4.2) | 84.5 ($\pm$ 7.2) | 52.0 ($\pm$ 11.6) |

Table 2: Average empirical reject option risk $\hat{\mathcal{R}}_\lambda(\Gamma_g)$ as defined in (1) for various values of $\lambda$. The best algorithms (smallest risk) as well as algorithms with non-significant differences are emphasized. The significance analysis uses a paired student t-test with level $\alpha = 0.05$.

| Algorithms | Average $\hat{\mathcal{R}}_\lambda(\Gamma_g)$ ($\pm$ std.) across all experimental settings | | | | |
|---|---|---|---|---|---|
| | $\lambda = 0.00$ | $\lambda = 0.05$ | $\lambda = 0.10$ | $\lambda = 0.15$ | $\lambda = 0.20$ |
| PL-KNN (2005) | 0.11 ($\pm$ 0.17) | 0.15 ($\pm$ 0.18) | 0.18 ($\pm$ 0.19) | 0.21 ($\pm$ 0.20) | 0.25 ($\pm$ 0.21) |
| PL-SVM (2008) | 0.19 ($\pm$ 0.27) | 0.23 ($\pm$ 0.28) | 0.27 ($\pm$ 0.29) | 0.31 ($\pm$ 0.30) | 0.35 ($\pm$ 0.31) |
| IPAL (2015) | 0.19 ($\pm$ 0.17) | 0.20 ($\pm$ 0.18) | 0.21 ($\pm$ 0.19) | 0.22 ($\pm$ 0.20) | 0.23 ($\pm$ 0.21) |
| PL-ECOC (2017) | 0.25 ($\pm$ 0.27) | 0.29 ($\pm$ 0.27) | 0.34 ($\pm$ 0.28) | 0.38 ($\pm$ 0.28) | 0.43 ($\pm$ 0.28) |
| PRODEN (2020) | 0.12 ($\pm$ 0.11) | 0.13 ($\pm$ 0.11) | 0.14 ($\pm$ 0.12) | 0.15 ($\pm$ 0.12) | **0.16** ($\pm$ 0.13) |
| CC (2020) | 0.16 ($\pm$ 0.18) | 0.16 ($\pm$ 0.19) | 0.17 ($\pm$ 0.20) | 0.18 ($\pm$ 0.20) | 0.19 ($\pm$ 0.21) |
| VALEN (2021) | 0.20 ($\pm$ 0.14) | 0.20 ($\pm$ 0.14) | 0.20 ($\pm$ 0.14) | 0.20 ($\pm$ 0.14) | 0.21 ($\pm$ 0.14) |
| POP (2023) | 0.12 ($\pm$ 0.11) | 0.13 ($\pm$ 0.11) | 0.14 ($\pm$ 0.12) | 0.15 ($\pm$ 0.12) | **0.16** ($\pm$ 0.13) |
| CROSEL (2024) | 0.15 ($\pm$ 0.20) | 0.16 ($\pm$ 0.21) | 0.17 ($\pm$ 0.21) | 0.18 ($\pm$ 0.21) | 0.19 ($\pm$ 0.22) |
| DST-PLL (ours) | **0.05** ($\pm$ 0.07) | **0.07** ($\pm$ 0.08) | **0.10** ($\pm$ 0.10) | **0.12** ($\pm$ 0.11) | **0.15** ($\pm$ 0.12) |

Table 3: Fraction of rejects and non-rejected test-set accuracy of all methods with standard deviations across all experimental settings and five repetitions with different random seeds. To obtain the results, we tune the thresholds $\theta$ such that each competitor has a number of rejects that is comparable to that of our method.

| Algorithms | Fraction of rejects ($\pm$ std.) | Non-rejected test accuracy ($\pm$ std.) |
|---|---|---|
| Pl-Knn (2005) | 50.19 % ($\pm$ 20.98 %) | 91.23 % ($\pm$ 10.12 %) |
| Pl-Svm (2008) | 50.19 % ($\pm$ 20.98 %) | 74.40 % ($\pm$ 19.77 %) |
| Ipal (2015) | 50.19 % ($\pm$ 20.98 %) | 83.52 % ($\pm$ 16.08 %) |
| Pl-Ecoc (2017) | 50.19 % ($\pm$ 20.98 %) | 73.92 % ($\pm$ 17.11 %) |
| Proden (2020) | 50.19 % ($\pm$ 20.98 %) | 94.03 % ($\pm$ 8.23 %) |
| Cc (2020) | 50.19 % ($\pm$ 20.98 %) | 90.13 % ($\pm$ 17.89 %) |
| Valen (2021) | 50.19 % ($\pm$ 20.98 %) | 86.81 % ($\pm$ 13.04 %) |
| Pop (2023) | 50.19 % ($\pm$ 20.98 %) | 94.02 % ($\pm$ 8.20 %) |
| CroSel (2024) | 50.19 % ($\pm$ 20.98 %) | 89.71 % ($\pm$ 18.67 %) |
| Dst-Pll (ours) | 50.15 % ($\pm$ 21.16 %) | **95.49 %** ($\pm$ 7.01 %) |

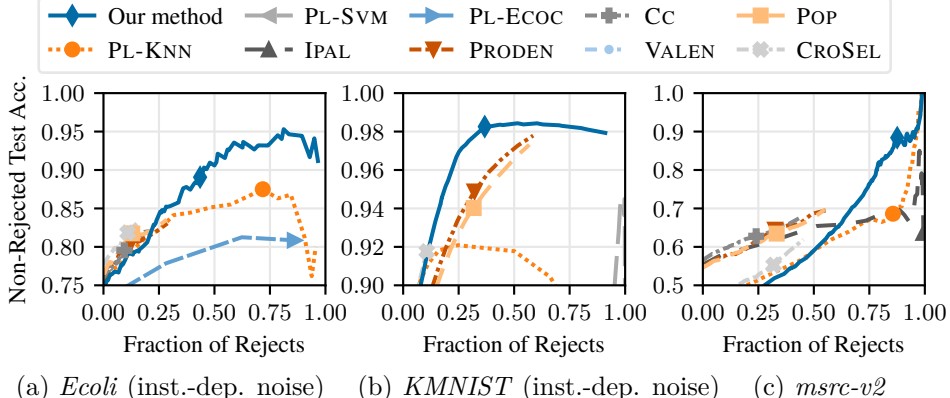

(a) *Ecoli* (inst.-dep. noise)  (b) *KMNIST* (inst.-dep. noise)  (c) *msrc-v2*

Figure 1: Trade-off between the fraction of rejected predictions and the accuracy of non-rejected predictions for three experiments: *Ecoli* with instance-dependent noise, *KMNIST* with instance-dependent noise, and the real-world dataset *msrc-v2*. We show the trade-off curves for varying confidence (0 to 1) and $\Delta_{\tilde{m}}$ (-1 to 1) thresholds. We highlight the points corresponding to a threshold of $\Delta_{\tilde{m}} = 0$ for our method, a confidence threshold of 90 % for methods with a probability output, and a threshold of 50 % of all votes for PL-KNN. We refer to Appendix D.4 for all reject trade-off curves across all experimental settings.

$\alpha = 0.05$ are emphasized. When misclassification is costly ($\lambda \leq 0.2$), our method provides the significantly best trade-off compared to our competitors. In contrast, when rejecting predictions is costly ($\lambda > 0.2$), the methods in Table 1 are to be preferred.

Table 3 shows the non-rejected test accuracy of all methods across all experimental settings for a fixed fraction of rejects. To obtain the results, we tune the confidence thresholds $\theta \in [0, 1]$ such that each competitor rejects a similar number of instances as our proposed approach. Our approach uses the threshold $\Delta > 0$, for which we have proved several desirable guarantees in Theorem 4.2. Our method achieves superior test-set accuracy on the non-rejected predictions.

Figure 1 shows the reject trade-off for varying confidence (0 to 1) and $\Delta_{\tilde{m}}$ (-1 to 1) thresholds on the *ecoli* and *KMNIST* datasets with instance-dependent noise as well as on the *msrc-v2* real-world dataset. The x-axes show the fractions of predictions that are rejected. The y-axes show the accuracies of predictions that are not rejected. The plots show (fraction of rejects, non-rejected test-set accuracy)-pairs corresponding to different settings of the thresholds. Most methods have monotonic growth: The more predictions are rejected, the more accurate are non-rejected predictions. Also, it is desirable to be close to the top-left corner of the plots as one wants to achieve high accuracy while rejecting as few predictions as possible. Note that the point $(1, 1)$ cannot be observed as the test-set accuracy is undefined if all predictions are rejected.

Given a desired rejection rate $\hat{r}(\Gamma_g)$, Figure 1 also allows for numerically finding the appropriate value of $\lambda$ that minimizes $\hat{\mathcal{R}}_\lambda(\Gamma_g)$. Varying $\lambda$ in (1), while having $\hat{er}(\Gamma_g)$ and $\hat{r}(\Gamma_g)$ fixed, yields a straight line in Figure 1 showing all possible trade-offs.

Our method provides the significantly best trade-offs compared to competitors (Table 2). These results empirically demonstrate our method's superiority in reliably rejecting unsure predictions.

## 6 Conclusions

When misclassification is costly, reject options provide a principled way of alleviating the consequences of incorrect predictions. In this work, we have presented a novel nearest-neighbor-based partial-label learning algorithm that can reject predictions if uncertain. We have demonstrated the desirable properties of the proposed reject option both from a theoretical (Theorem 4.2) and practical perspective. Our wide range of experiments showed the effectiveness of our classification rule and reject option on supervised datasets with added artificial noise and partially labeled real-world datasets.

**Acknowledgments**

We thank the anonymous reviewers for their constructive comments. This work was supported by the German Research Foundation (DFG) Research Training Group GRK 2153: *Energy Status Data — Informatics Methods for its Collection, Analysis and Exploitation* and by the KiKIT (The Pilot Program for Core-Informatics at the KIT) of the Helmholtz Association.

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

# A  Proofs

This section collects our proofs. Appendix A.1 contains the proof of Theorem 4.2. The proof of Lemma 4.5 is in Appendix A.2. We prove our main statement, Theorem 4.6, in Appendix A.3.

## A.1  Proof of Theorem 4.2

**Part** $(i)$**.** Given an instance $\tilde{x} \in \mathcal{X}$ with its candidate set $\tilde{s} \subseteq \mathcal{Y}$ and its associated prediction $g(\tilde{x})$ as described in Algorithm 1, we assume that $g(\tilde{x})$ has been picked randomly from $\arg\max_{A \subseteq \tilde{s}} \tilde{m}(A)$ (Algorithm 1, Line 11, second case), so $\max_{y \in \tilde{s}} \tilde{m}(\{y\})$ must be zero because we are not in the first case in Line 11. Therefore, $\mathrm{bel}_{\tilde{m}}(\{y\}) \overset{(2)}{=} \tilde{m}(\{y\}) = 0$ for all $y \in \mathcal{Y}$. Substitution in $\Delta_{\tilde{m}}$ yields

$$\Delta_{\tilde{m}} = \mathrm{bel}_{\tilde{m}}(\{y\}) - \max_{y' \in \tilde{s} \setminus \{y\}} \mathrm{pl}_{\tilde{m}}(\{y'\}) = 0 - \max_{y' \in \tilde{s} \setminus \{y\}} \mathrm{pl}_{\tilde{m}}(\{y'\}) \leq 0,$$

independent of the choice of $y \in \mathcal{Y}$. Therefore, $\Delta_{\tilde{m}} \leq 0$.

**Part** $(ii)$**.** Given an instance $\tilde{x} \in \mathcal{X}$ with its candidate set $\tilde{s} \subseteq \mathcal{Y}$ and its associated prediction $g(\tilde{x})$ as described in Algorithm 1, we assume that $\Delta_{\tilde{m}} > 0$. By the contraposition of part $(i)$, $g(\tilde{x}) = \hat{y}$ with $\hat{y} = \arg\max_{y \in \tilde{s}} \tilde{m}(\{y\})$. From (5) and $\Delta_{\tilde{m}} > 0$, it follows that $\mathrm{bel}_{\tilde{m}}(\{\hat{y}\}) > \max_{y \in \tilde{s} \setminus \{\hat{y}\}} \mathrm{pl}_{\tilde{m}}(\{y\})$, which is equivalent to $\mathrm{bel}_{\tilde{m}}(\{\hat{y}\}) > \mathrm{pl}_{\tilde{m}}(\{y\})$ for all $y \in \tilde{s} \setminus \{\hat{y}\}$. For all $\mathbb{P} \in \mathcal{C}_{\tilde{m}}(\mathcal{Y}, 2^{\mathcal{Y}})$, it holds that $\mathrm{bel}_{\tilde{m}}(A) \leq \mathbb{P}(A) \leq \mathrm{pl}_{\tilde{m}}(A)$ for all $A \subseteq \mathcal{Y}$ by (3). Therefore, $\mathbb{P}(\{y\}) \leq \mathrm{pl}_{\tilde{m}}(\{y\}) < \mathrm{bel}_{\tilde{m}}(\{\hat{y}\}) \leq \mathbb{P}(\{\hat{y}\})$ for all $y \in \tilde{s} \setminus \{\hat{y}\}$.

**Part** $(iii)$**.** $(\Rightarrow)$: Given an instance $\tilde{x} \in \mathcal{X}$ with its candidate set $\tilde{s} \subseteq \mathcal{Y}$ and its associated prediction $g(\tilde{x})$ as described in Algorithm 1, we assume that $\tilde{m}(\{g(\tilde{x})\}) > 1/2$. Because $\max_{y \in \tilde{s}} \tilde{m}(\{y\}) > 0$, we are in the first case in Line 11 of Algorithm 1. Therefore, $g(\tilde{x}) = \hat{y}$ with $\hat{y} = \arg\max_{y \in \tilde{s}} \tilde{m}(\{y\})$. As $\sum_{A \subseteq \mathcal{Y}} \tilde{m}(A) = 1$ and $\tilde{m}(\{\hat{y}\}) > 1/2$, it holds that $\sum_{A \subseteq \mathcal{Y}, A \neq \{\hat{y}\}} \tilde{m}(A) < 1/2$. Then, for all $y \in \tilde{s}$ with $y \neq \hat{y}$, $\mathrm{pl}_{\tilde{m}}(\{y\}) = \sum_{A \subseteq \mathcal{Y}, A \cap \{y\} \neq \emptyset} \tilde{m}(A) < 1/2$. Hence, $\mathrm{pl}_{\tilde{m}}(\{y\}) < \mathrm{bel}_{\tilde{m}}(\{\hat{y}\})$ for all $y \in \tilde{s}$ with $y \neq \hat{y}$. Therefore, $\Delta_{\tilde{m}} = \mathrm{bel}_{\tilde{m}}(\{\hat{y}\}) - \max_{y \in \tilde{s} \setminus \{\hat{y}\}} \mathrm{pl}_{\tilde{m}}(\{y\}) > 0$.

$(\not\Leftarrow)$: In the following, we provide a counter-example. Let $\tilde{s} = \mathcal{Y} = \{1, 2, 3\}$ and $\tilde{m}$ be defined by $\tilde{m}(A) = 0.4$ if $A = \{1\}$, $\tilde{m}(A) = 0.3$ if $A = \{1, 2\}$ or $A = \{1, 3\}$, else $\tilde{m}(A) = 0$. Then, $y = 1$ is our prediction since it has the highest probability mass. The prediction is not rejected because $\Delta_{\tilde{m}} = 0.4 - 0.3 = 0.1 > 0$. However, $\tilde{m}(\{y\}) < 1/2$.

## A.2  Proof of Lemma 4.5

**Part** $(i)$**.** The expected probability mass of $\tilde{y}$ is given by (7). The non-negativity of the binomial coefficients implies that it is sufficient to show that

$$\binom{k-h}{i} \prod_{a=1}^{l-2} \binom{k-h}{j_a} \geq \sum_{b=1}^{\min(i,j_1,\dots,j_{l-2})} (-1)^{b+1} \binom{k-h}{b} \binom{k-h-b}{i-b} \prod_{c=1}^{l-2} \binom{k-h-b}{j_c-b}, \tag{6}$$

where $k \in \mathbb{N}$, $h, i, j_a \in \mathbb{N}_0$ with $0 \leq i < k-h$ and $0 \leq j_a < k-h$ with $a \in [l-2]$, and that the inequality is strict in at least one case.

The proof of (6) proceeds by induction over $k$. Note that the choices of $i$ and $j_a$ are interchangeable (see Appendix B). Hence, w.l.o.g., we assume for the remainder of this proof that $i \leq j_a$ for all $a \in [l-2]$. Further, we set $k' = k - h \in \mathbb{N}$.

*Proposition.* For $k' \in \mathbb{N}$, it holds that

$$\binom{k'}{i} \prod_{a=1}^{l-2} \binom{k'}{j_a} \geq \sum_{b=1}^{i} (-1)^{b+1} \binom{k'}{b} \binom{k'-b}{i-b} \prod_{c=1}^{l-2} \binom{k'-b}{j_c-b},$$

with $0 \leq i \leq j_a < k'$ for all $a \in [l-2]$.

*Base case.* For $k' = 1$, it holds that $h = i = j_a = 0$ ($a \in [l-2]$). Then, all binomial coefficients on the l.h.s. are 1. The sum on the r.h.s. is empty and thus 0. Hence, l.h.s. > r.h.s. ($1 > 0$), which also renders (7) strictly positive.

*Induction step.* Let $k' = k' + 1$. The l.h.s. yields

$$
\binom{k'+1}{i} \prod_{a=1}^{l-2} \binom{k'+1}{j_a} \overset{(i)}{=} \left( \frac{k'+1}{k'+1-i} \prod_{a=1}^{l-2} \frac{k'+1}{k'+1-j_a} \right) \binom{k'}{i} \prod_{a=1}^{l-2} \binom{k'}{j_a}
$$

$$
\overset{(ii)}{\geq} \left( \frac{k'+1}{k'+1-i} \prod_{a=1}^{l-2} \frac{k'+1}{k'+1-j_a} \right) \sum_{b=1}^{i} (-1)^{b+1} \binom{k'}{b} \binom{k'-b}{i-b} \prod_{c=1}^{l-2} \binom{k'-b}{j_c - b}
$$

$$
\overset{(iii)}{=} \sum_{b=1}^{i} (-1)^{b+1} \frac{k'+1}{k'+1-i} \binom{k'}{b} \binom{k'-b}{i-b} \prod_{c=1}^{l-2} \frac{k'+1}{k'+1-j_c} \binom{k'-b}{j_c - b}
$$

$$
\overset{(iv)}{=} \sum_{b=1}^{i} (-1)^{b+1} \underbrace{\frac{k'+1}{k'+1-b} \binom{k'}{b}}_{\overset{(i')}{=} \binom{k'+1}{b}} \underbrace{\frac{k'+1-b}{k'+1-i} \binom{k'-b}{i-b}}_{\overset{(i')}{=} \binom{k'+1-b}{i-b}} \prod_{c=1}^{l-2} \frac{k'+1}{k'+1-b} \underbrace{\frac{k'+1-b}{k'+1-j_c} \binom{k'-b}{j_c - b}}_{\overset{(i')}{=} \binom{k'+1-b}{j_c - b}}
$$

$$
\overset{(v)}{=} \sum_{b=1}^{i} (-1)^{b+1} \left( \frac{k'+1}{k'+1-b} \right)^{l-2} \binom{k'+1}{b} \binom{k'+1-b}{i-b} \prod_{c=1}^{l-2} \binom{k'+1-b}{j_c - b}
$$

$$
\overset{(vi)}{\geq} \sum_{b=1}^{i} (-1)^{b+1} \binom{k'+1}{b} \binom{k'+1-b}{i-b} \prod_{c=1}^{l-2} \binom{k'+1-b}{j_c - b},
$$

where $(i)$ holds as

$$
\binom{k'+1}{i} = \frac{(k'+1)!}{i!(k'+1-i)!} = \frac{k'+1}{k'+1-i} \frac{k'!}{i!(k'-i)!} = \frac{k'+1}{k'+1-i} \binom{k'}{i},
$$

the proposition yields $(ii)$, in $(iii)$ we rearrange the sum and collect both products, and $(iv)$ expands the fractions. In $(v)$, we rearrange the factor within the product and apply the $(i')$-s, which hold by similar calculations as $(i)$, and Lemma B.1 with $k = k' + 1$ implies $(vi)$. These derivations and the observation that the base case guarantees the inequality to be strict conclude the proof.

**Part** $(ii)$**.** By comparing the closed-form expressions (with $m = i = t$) of $\tilde{y}$ and $\tilde{y}_c$ given in (7) and (8), respectively, we obtain

$$
\mathbb{E}_{\mathbb{P}}[\tilde{m}(\{\tilde{y}\}) \mid X = \tilde{x}] > \mathbb{E}_{\mathbb{P}}[\tilde{m}(\{\tilde{y}_c\}) \mid X = \tilde{x}]
$$

$$
\Leftrightarrow \left( \frac{1}{2^{l-2}} p_2 \right)^{k-h-m} > \left( \frac{1}{2^{l-1}-1} p_3 \right)^{k-h-m}
$$

$$
\Leftrightarrow \frac{1}{2^{l-2}} p_2 > \frac{1}{2^{l-1}-1} p_3 \overset{(i)}{\Leftrightarrow} \frac{2^{l-1}-1}{2^{l-2}} > 1 \Leftrightarrow 2^{l-2} > 1,
$$

where $(i)$ holds as Assumption 4.4 guarantees that $p_2 \geq p_3 > 0$. The last statement is satisfied for $l \geq 3$, which shows that the dominance assumption w.r.t. the label distribution propagates through the computation of belief when using Yager's rule. We note that the result is independent of $p_1$ as a candidate set can not distinguish the belief in $\tilde{y}$ and $\tilde{y}_c$ when these co-occur.

### A.3 Proof of Theorem 4.6

With Assumption 4.4, Theorem 4.6 is a consequence of Lemma 4.5. We detail this in the following.

**Part** $(i)$**.** Let $\tilde{x} \in \mathcal{X}$ be a fixed instance with unknown true label $\tilde{y}$. Our algorithm is trained with $n$ samples $((x_i, s_i))_{i=1}^{n} \overset{\text{i.i.d.}}{\sim} \mathbb{P}_{XS}$ that satisfy the label distribution in Assumption 4.4. Especially, the dominance of the

true label $\tilde{y}$ within $\tilde{x}$'s neighborhood of $k$ instances $x_i$ holds. Since the expected probability mass of $\tilde{y}$ is positive (Lemma 4.5; part $(i)$), that is, $\mathbb{E}_{\mathbb{P}}[\tilde{m}(\{\tilde{y}\}) \mid X = \tilde{x}] > 0$, we are, in expectation, in the first case of our two-case decision rule (Algorithm 1, Line 11). Furthermore, label $\tilde{y}$ receives, in expectation, the maximum probability mass among all labels because the dominance of label $\tilde{y}$ in the neighborhood propagates to the label masses by Lemma 4.5 (part $(ii)$), that is, $\mathbb{E}_{\mathbb{P}}[\tilde{m}(\{\tilde{y}\}) \mid X = \tilde{x}] > \mathbb{E}_{\mathbb{P}}[\tilde{m}(\{\tilde{y}_{\mathrm{c}}\}) \mid X = \tilde{x}]$. Therefore, in expectation, it must hold that, given an instance $\tilde{x}$, we consistently predict the true label $\tilde{y}$, that is, $\mathbb{E}_{((x_i,s_i))_{i=1}^n \overset{\text{i.i.d.}}{\sim} \mathbb{P}_{XS}} g_n(\tilde{x}) = \tilde{y}$.

**Part** $(ii)$. Given $n$ samples $((x_i, s_i))_{i=1}^n \overset{\text{i.i.d.}}{\sim} \mathbb{P}_{XS}$, the classifier $g_n$ trained by Algorithm 1 from the $n$ samples, and the minimum loss $\mathcal{L}(g(x), s) = \min_{y \in s} \mathbb{1}_{\{g(x) \neq y\}}$ (Lv et al., 2020; Feng et al., 2020), the expected empirical risk is

$$\mathbb{E}_{\mathbb{P}}[\hat{\mathcal{R}}(g_n)] = \mathbb{E}_{\mathbb{P}}\left[\frac{1}{n}\sum_{i=1}^n \mathcal{L}(g_n(x_i), s_i)\right] = \frac{1}{n}\sum_{i=1}^n \mathbb{E}_{\mathbb{P}}[\mathcal{L}(g_n(x_i), s_i)] \overset{(i)}{=} 0,$$

which is zero by part $(i)$ of the statement as, in expectation, we consistently predict the true label $y_i$ for any fixed label $x_i$.

By the law of large numbers, we have

$$\lim_{n \to \infty} \hat{\mathcal{R}}(g_n) = \lim_{n \to \infty} \frac{1}{n}\sum_{i=1}^n \mathcal{L}(g_n(x_i), s_i) = 0 \ \text{ almost surely,}$$

because $\mathbb{E}_{\mathbb{P}}[\hat{\mathcal{R}}(g)] = 0$ as stated above. As $\mathcal{R}(g^*) \geq 0$, we conclude

$$0 \leq \lim_{n \to \infty} \left(\hat{\mathcal{R}}(g_n) - \mathcal{R}(g^*)\right) \leq \lim_{n \to \infty} \hat{\mathcal{R}}(g_n) = 0 \ \text{ almost surely,}$$

which establishes $(ii)$.

# B   Auxiliary Results

This section lists results underpinning the findings in Appendix A. Section B.1 introduces a closed-form expression of the expected probability mass. Section B.2 presents Lemma B.1, which is needed in the proof of Lemma 4.5.

## B.1   Expected Probability Mass

This section, first, presents a closed-form expression for the expected probability mass of $\tilde{y}$ and $\tilde{y}_{\mathrm{c}}$. Second, we compare the computation with a simulation to numerically validate our closed-form expression and the results in Appendix A.

**Closed-form expression.** Let $\tilde{x} \in \mathcal{X}$ be the instance of interest with hidden true label $\tilde{y} \in \mathcal{Y}$ and $k$ associated neighbors $(x_i, s_i) \in \mathrm{NN}_k(\tilde{x})$. We recall that our label space $\mathcal{Y}$ has $l \geq 3$ classes. Then, the expected

probability mass of $\tilde{y}$ given instance $\tilde{x}$ is

$$\mathbb{E}_{(X,Y,S)\sim\mathbb{P}}\big[\tilde{m}(\{\tilde{y}\}) \mid X = \tilde{x}\big] \overset{(i)}{=} \mathbb{E}_{(X,Y,S)\sim\mathbb{P}}\big[q(\{\tilde{y}\}) \mid X = \tilde{x}\big]$$

$$\overset{(ii)}{=} \mathbb{E}_{(X,Y,S)\sim\mathbb{P}}\Big[\sum_{\substack{A_1,\ldots,A_k\subseteq\mathcal{Y}\\ \bigcap_{i=1}^{k} A_i=\{\tilde{y}\}}} \prod_{j=1}^{k} m_j(A_j) \mid X = \tilde{x}\Big]$$

$$\overset{(iii)}{=} \sum_{h=0}^{k-1}\sum_{i=0}^{k-h-1}\sum_{j_1=0}^{k-h-1}\sum_{j_2=0}^{k-h-1}\cdots\sum_{j_{l-2}=0}^{k-h-1} \underbrace{\sum_{\substack{A_1,\ldots,A_k\subseteq\mathcal{Y};\\ B_1,\ldots,B_l\subseteq[k]\\ \text{with } |B_1|=h,|B_2|=i,\\ |B_{q+2}|=j_q \text{ for } q\in[l-2];\\ \tilde{y}\in A_r \text{ for } r\in[k];\\ A_r=\mathcal{Y} \text{ for } r\in B_1;\\ \tilde{y}_c\in A_r \text{ for } r\in B_2;\\ \tilde{y}_{q+2}\in A_{r_q} \text{ for } r_q\in B_{q+2}\\ \text{and } q\in[l-2]}}}_{(iv)} \underbrace{\Big(\prod_{j=1}^{k} m_j(A_j)\Big)}_{(v)} \underbrace{\Big(\prod_{j=1}^{k}\mathbb{P}(S=A_j, Y=\tilde{y}\mid X=x_j)\Big)}_{(vi)}$$

$$= \sum_{h=0}^{k-1}\sum_{i=0}^{k-h-1}\sum_{j_1=0}^{k-h-1}\sum_{j_2=0}^{k-h-1}\cdots\sum_{j_{l-2}=0}^{k-h-1} \underbrace{\sum_{\substack{A_1,\ldots,A_k\subseteq\mathcal{Y};\\ B_1,\ldots,B_l\subseteq[k]\\ \text{with } |B_1|=h,|B_2|=i,\\ |B_{q+2}|=j_q \text{ for } q\in[l-2];\\ \tilde{y}\in A_r \text{ for } r\in[k];\\ A_r=\mathcal{Y} \text{ for } r\in B_1;\\ \tilde{y}_c\in A_r \text{ for } r\in B_2;\\ \tilde{y}_{q+2}\in A_{r_q} \text{ for } r_q\in B_{q+2}\\ \text{and } q\in[l-2]}}}_{(vii)} \underbrace{\frac{1}{2^k}}_{(vi)} \underbrace{\Big(\frac{1}{2^{l-2}-1}p_1\Big)^{i}\Big(\frac{1}{2^{l-2}}p_2\Big)^{k-h-i}}_{(vi)}$$

$$= \sum_{h=0}^{k-1}\sum_{i=0}^{k-h-1}\sum_{j_1=0}^{k-h-1}\sum_{j_2=0}^{k-h-1}\cdots\sum_{j_{l-2}=0}^{k-h-1} \Big[\underbrace{\binom{k}{k-h}\binom{k-h}{i}\prod_{a=1}^{l-2}\binom{k-h}{j_a}}_{(vii)}$$

$$-\underbrace{\binom{k}{k-h}\sum_{b=1}^{\min(i,j_1,j_2,\ldots,j_{l-2})}(-1)^{b+1}\binom{k-h}{b}\binom{k-h-b}{i-b}\prod_{c=1}^{l-2}\binom{k-h-b}{j_c-b}}_{(vii)}\Big]$$

$$\times\frac{1}{2^k}\Big(\frac{1}{2^{l-2}-1}p_1\Big)^{i}\Big(\frac{1}{2^{l-2}}p_2\Big)^{k-h-i}. \tag{7}$$

In the following, we show that transformations $(i) - (viii)$ hold.

- In $(i)$, we apply (4). For any $y \in \mathcal{Y}$, it holds that $m(\{y\}) = q(\{y\})$ as $\{y\} \neq \emptyset$ and $\{y\} \neq \mathcal{Y}$.

- In $(ii)$, we insert (4), that is, how all $m_i$-s as defined by Algorithm 1 are combined into q using Yager's rule (4).

- In $(iii)$, we then compute the expected value by taking into account all possible combinations of sets $A_1, \ldots, A_k \subseteq \mathcal{Y}$ producing the intersection $\{\tilde{y}\}$.

- Term $(iv)$ enumerates all cases where $\bigcap_{r=1}^{k} A_r = \{\tilde{y}\}$. To produce this intersection, label $\tilde{y}$ needs to be contained in all sets $A_r$. At most $k - 1$ sets can be the full label space, that is, $h$ sets satisfy $A_r = \mathcal{Y}$. The label $\tilde{y}_c$ with which $\tilde{y}$ is most often confused has to be missing in at least one set $A_r$, that is, $i$ sets $A_r$ satisfy $\tilde{y}_c \in A_r$. All remaining labels $\tilde{y}_3, \ldots, \tilde{y}_l \in \mathcal{Y}$ also have to be missing in at least one set $A_r$ each, that is, $j_a$ sets $A_r$ satisfy $\tilde{y}_{a+2} \in A_r$, respectively.

- Term $(v)$ calculates the value that a given instantiation of all $A_1, \dots, A_k$ has. Because we only sum over focal elements of the $\mathrm{m}_j$-s $(j \in [k])$ by the construction of the sum in $(v)$, term $(vi)$ simplifies to $1/2^k$. Recall that all focal sets have a value of $1/2$ (Algorithm 1, Line 6).

- Expression $(vi)$ computes the probability that a given instantiation of all $A_1, \dots, A_k$ has. The variables $h$, $i$, $j_1$, $\dots$, $j_{l-2}$ directly allow to quantify the probabilities in a way to apply Assumption 4.4, which allows to simplify the expression.

- As the summands within $(vii)$ do not depend on the $A_1, \dots, A_k$ anymore, we can simplify by counting their possible instantiations. We first count the possible combinations of sets where we pick to intersect $h$-times with the full label space and $(k-h)$-times with the neighbors' candidate sets of which $i$ contain label $\tilde{y}_c$ and $j_a$ contain the remaining labels for all $a \in [l-2]$. Second, we need to subtract all combinations that produce the whole label space within at least one candidate set as the probability of $s' = \mathcal{Y}$ is zero per Assumption 4.4. We do this using an inclusion-exclusion strategy. We have $k-h$ sets with which we intersect and that are not the full label space. Out of those $k-h$ sets, at least one set ought to contain all possible labels leaving $\binom{k-h-1}{i-1}$ and $\binom{k-h-1}{j_c-1}$ possibilities for the other sets. This overcounts combinations though as multiple sets can contain all labels. We employ the inclusion-exclusion strategy to remove combinations that produce the same allocations of labels to sets.

Similar to (7), we also express the expected mass of the most frequently co-occurring label $\tilde{y}_c$ as

$$\mathbb{E}_{(X,Y,S)\sim\mathbb{P}}\big[\tilde{\mathrm{m}}(\{\tilde{y}_c\}) \mid X = \tilde{x}\big] = \mathbb{E}_{(X,Y,S)\sim\mathbb{P}}\big[\mathrm{q}(\{\tilde{y}_c\}) \mid X = \tilde{x}\big]$$

$$= \mathbb{E}_{(X,Y,S)\sim\mathbb{P}}\Big[ \sum_{\substack{A_1,\dots,A_k \subseteq \mathcal{Y} \\ \bigcap_{i=1}^k A_i = \{\tilde{y}_c\}}} \prod_{j=1}^k \mathrm{m}_j(A_j) \mid X = \tilde{x}\Big]$$

$$= \sum_{h=0}^{k-1} \sum_{t=0}^{k-h-1} \sum_{j_1=0}^{k-h-1} \sum_{j_2=0}^{k-h-1} \cdots \sum_{j_{l-2}=0}^{k-h-1} \Bigg[ \binom{k}{k-h}\binom{k-h}{t} \prod_{a=1}^{l-2}\binom{k-h}{j_a}\Bigg.$$

$$\Bigg. - \binom{k}{k-h} \sum_{b=1}^{\min(t,j_1,j_2,\dots,j_{l-2})} (-1)^{b+1}\binom{k-h}{b}\binom{k-h-b}{t-b}\prod_{c=1}^{l-2}\binom{k-h-b}{j_c-b}\Bigg]$$

$$\times \frac{1}{2^k}\left(\frac{1}{2^{l-2}-1}p_1\right)^t \left(\frac{1}{2^{l-1}-1}p_3\right)^{k-h-t}. \tag{8}$$

To produce the intersection $\{\tilde{y}_c\}$ in Yager's rule, all $k-h$ sets with which we intersect need to contain $\tilde{y}_c$. The variable $t$ denotes how many out of $k-h$ sets contain the true label $\tilde{y}$. Therefore, $\tilde{y}$ and $\tilde{y}_c$ co-occur in a set in $t$ cases, with which the probability $(\frac{1}{2^{l-2}-1}p_1)^t$ is associated. In $k-h-t$ sets, the true label is not contained, with which the probability $(\frac{1}{2^{l-1}-1}p_3)^{k-h-t}$ is associated according to Assumption 4.4.

**Simulation.** To simulate the expected probability masses, we randomly draw $k$ candidate sets $s_i$ according to the label distribution in Assumption 4.4, apply Algorithm 1, and report the belief of $\tilde{y}$ and $\tilde{y}_c$. We repeat this $100\,000$ times and average the results. For $l = 3$, $\tilde{y} = 1$, and $\tilde{y}_c = 2$, we obtain the label distribution

$$\mathbb{P}(S = \{1,2\}, Y = 1 \mid X = x_i) = \quad p_1,$$

$$\mathbb{P}(S = \{1\}, Y = 1 \mid X = x_i) = \mathbb{P}(S = \{1,3\}, Y = 1 \mid X = x_i) = \frac{1}{2}p_2,$$

$$\mathbb{P}(S = \{2\}, Y = 2 \mid X = x_i) = \mathbb{P}(S = \{3\}, Y = 3 \mid X = x_i) = \frac{1}{3}p_3,$$

$$\mathbb{P}(S = \{2,3\}, Y = 2 \mid X = x_i) = \mathbb{P}(S = \{2,3\}, Y = 3 \mid X = x_i) = \frac{1}{6}p_3,$$

regarding a neighboring instance $x_i$ with $i \in [k]$. All other probabilities that are not listed are zero. Naturally, $\sum_{s' \subseteq \mathcal{Y}} \sum_{y' \in \mathcal{Y}} \mathbb{P}(S = s', Y = y' \mid X = x_i) = p_1 + 2\frac{1}{2}p_2 + 2\frac{1}{3}p_3 + 2\frac{1}{6}p_3 = 1$.

Figure 2 compares the computation and simulation of the expected probability masses. The values of the computation and simulation are indiscernible. Note that the expected probability mass of any single-item set converges towards zero as $k$ increases. The more sets one intersects with in (4), the more likely it is to produce the empty set. As shown in Lemma 4.5 (part $(ii)$), the expected probability mass of $\tilde{y}$ is greater than the mass of $\tilde{y}_c$, independently from the number of neighbors $k \in \mathbb{N}$.

### B.2 Auxiliary Lemma to the Proof of Lemma 4.5

This section presents an auxiliary inequality, which is used in the proof of Lemma 4.5.

**Lemma B.1.** *Let $l \in \mathbb{N}$ with $l \geq 3$ denote the number of class labels and $k \in \mathbb{N}$ the number of neighbors. Let further $i, j_a \in \mathbb{N}_0$ with $0 \leq i \leq j_a < k$ for all $a \in [l-2]$. Then, it holds that*

$$\sum_{b=1}^{i} (-1)^{b+1} \left(\frac{k}{k-b}\right)^{l-2} \binom{k}{b}\binom{k-b}{i-b}\prod_{c=1}^{l-2}\binom{k-b}{j_c-b} \geq \sum_{b=1}^{i}(-1)^{b+1}\binom{k}{b}\binom{k-b}{i-b}\prod_{c=1}^{l-2}\binom{k-b}{j_c-b}.$$

*Proof.* We first transform the statement as

$$\sum_{b=1}^{i} (-1)^{b+1} \left(\frac{k}{k-b}\right)^{l-2} \binom{k}{b}\binom{k-b}{i-b}\prod_{c=1}^{l-2}\binom{k-b}{j_c-b} \geq \sum_{b=1}^{i}(-1)^{b+1}\binom{k}{b}\binom{k-b}{i-b}\prod_{c=1}^{l-2}\binom{k-b}{j_c-b}$$

$$\Leftrightarrow \sum_{b=1}^{i} (-1)^{b+1} \left(\left(\frac{k}{k-b}\right)^{l-2}-1\right) \binom{k}{b}\binom{k-b}{i-b}\prod_{c=1}^{l-2}\binom{k-b}{j_c-b} \geq 0.$$

We then write the latter as

$$\sum_{b=1}^{i} (-1)^{b+1} \underbrace{\left(\left(\frac{k}{k-b}\right)^{l-2}-1\right)}_{(i)} \binom{k}{b}\binom{k-b}{i-b}\prod_{c=1}^{l-2}\binom{k-b}{j_c-b}$$

$$= \sum_{b=1}^{i} (-1)^{b+1} \left(\sum_{c=0}^{l-3}\left(\frac{k}{k-b}\right)^{c}\right)\frac{b}{k-b}\binom{k}{b}\binom{k-b}{i-b}\underbrace{\prod_{c=1}^{l-2}\binom{k-b}{j_c-b}}_{(ii)}$$

$$= \sum_{b=1}^{i} (-1)^{b+1} \left(\sum_{c=0}^{l-3}\left(\frac{k}{k-b}\right)^{c}\right)\underbrace{\frac{b}{k-b}\binom{k}{b}\binom{k-b}{i-b}\binom{k-b}{j_1-b}}_{(iii)}\prod_{c=2}^{l-2}\binom{k-b}{j_c-b}$$

$$= \sum_{b=1}^{i} (-1)^{b+1} \left(\sum_{c=0}^{l-3}\left(\frac{k}{k-b}\right)^{c}\right)\frac{i}{k-j_1}\binom{k}{i}\binom{i-1}{b-1}\binom{k-b-1}{j_1-b}\prod_{c=2}^{l-2}\binom{k-b}{j_c-b}$$

$$\overset{(iv)}{=} \frac{i}{k-j_1}\binom{k}{i}\sum_{b=1}^{i}(-1)^{b+1}\left(\sum_{c=0}^{l-3}\left(\frac{k}{k-b}\right)^{c}\right)\binom{i-1}{b-1}\binom{k-b-1}{j_1-b}\prod_{c=2}^{l-2}\binom{k-b}{j_c-b}$$

$$\overset{(v)}{=} \frac{i}{k-j_1}\binom{k}{i}\sum_{b=0}^{i-1}(-1)^{b}\left(\sum_{c=0}^{l-3}\left(\frac{k}{k-b-1}\right)^{c}\right)\underbrace{\binom{i-1}{b}\binom{k-b-2}{j_1-b-1}}_{(vi)}\prod_{c=2}^{l-2}\binom{k-b-1}{j_c-b-1}$$

$$= \frac{i}{k-j_1}\binom{k}{i}\sum_{b=0}^{i-1}(-1)^{b}\left(\sum_{c=0}^{l-3}\left(\frac{k}{k-b-1}\right)^{c}\right)(-1)^{-b}\binom{k-i-1}{j_1-1}\prod_{c=2}^{l-2}\binom{k-b-1}{j_c-b-1}$$

$$\overset{(vii)}{=} \frac{i}{k-j_1}\binom{k}{i}\binom{k-i-1}{j_1-1}\sum_{b=0}^{i-1}\left(\sum_{c=0}^{l-3}\left(\frac{k}{k-b-1}\right)^{c}\right)\prod_{c=2}^{l-2}\binom{k-b-1}{j_c-b-1} \geq 0,$$

which shows the statement to be demonstrated as the last term is a sum of products whose factors are all non-negative. Recall that $0 \leq i \leq j_a < k$ for all $a \in [l-2]$. In the following, we show that all transformations are sound.

- Using Theorem C.2, $(i)$ holds as

$$
\left(\frac{k}{k-b}\right)^{l-2} - 1 = \left(\left(\frac{k}{k-b}\right)^{l-2} - \left(\frac{k}{k-b}\right)^{l-3}\right) + \left(\left(\frac{k}{k-b}\right)^{l-3} - \left(\frac{k}{k-b}\right)^{l-4}\right)
$$
$$
+ \cdots + \left(\frac{k}{k-b} - 1\right)
$$
$$
= \left(\frac{k}{k-b} - 1\right)\left(\left(\frac{k}{k-b}\right)^{l-3} + \left(\frac{k}{k-b}\right)^{l-4} + \cdots + 1\right)
$$
$$
= \left(\frac{k}{k-b} - 1\right) \sum_{c=0}^{l-3} \left(\frac{k}{k-b}\right)^{c} = \frac{b}{k-b} \sum_{c=0}^{l-3} \left(\frac{k}{k-b}\right)^{c}.
$$

- In $(ii)$, we extract the first factor of the product such that it starts with $c = 2$.

- Subsequently, we transform the binomial coefficients $(iii)$ as

$$
\frac{b}{k-b}\binom{k}{b}\binom{k-b}{i-b}\binom{k-b}{j_1-b}
$$
$$
\overset{(C.4)}{=} \frac{b}{k-b}\binom{k}{i}\binom{i}{b}\binom{k-b}{j_1-b}
$$
$$
\overset{(C.1)}{=} \frac{b}{k-b}\frac{i!}{b!\,(i-b)!}\frac{(k-b)!}{(j_1-b)!\,(k-j_1)!}\binom{k}{i}
$$
$$
\overset{(viii)}{=} \frac{i}{k-j_1}\frac{(i-1)!}{(b-1)!\,(i-b)!}\frac{(k-b-1)!}{(j_1-b)!\,(k-j_1-1)!}\binom{k}{i}
$$
$$
\overset{(C.1)}{=} \frac{i}{k-j_1}\binom{k}{i}\binom{i-1}{b-1}\binom{k-b-1}{j_1-b},
$$

  where $(viii)$ rearranges terms.

- Then, in $(iv)$, we move terms that are independent of $b$ outside of the summation.

- In $(v)$, we shift the summation to start at $b = 0$.

- Finally, by defining $r = j_1 - b - 1 \in \mathbb{N}_0$, we express $(vi)$ as

$$
\binom{i-1}{b}\binom{k-b-2}{j_1-b-1} = \binom{i-1}{b}\binom{r+k-j_1-1}{r}
$$
$$
\overset{(C.3)}{=} (-1)^r \binom{i-1}{b}\binom{j_1-k}{r}
$$
$$
\overset{(ix)}{=} (-1)^r \sum_{\substack{b',r' \in \mathbb{N}_0 \\ b'+r'=j_1-1}} \binom{i-1}{b'}\binom{j_1-k}{r'}
$$
$$
\overset{(C.5)}{=} (-1)^{j_1-b-1}\binom{i+j_1-k-1}{j_1-1}
$$
$$
\overset{(C.3)}{=} (-1)^{j_1-b-1}(-1)^{j_1-1}\binom{k-i-1}{j_1-1}
$$
$$
= (-1)^{-b}\binom{k-i-1}{j_1-1}.
$$

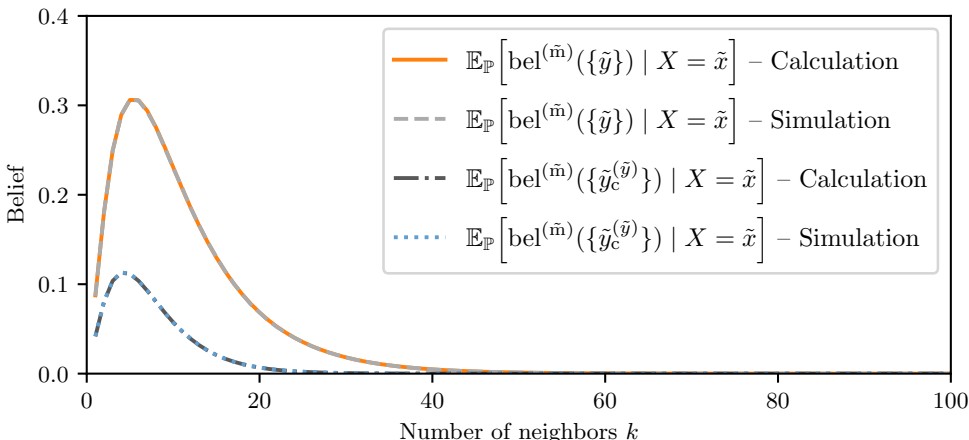

Figure 2: Comparison of simulating and calculating the expected belief of the true label $\tilde{y}$ and of the most frequently co-occurring label $\tilde{y}_c$ for $k \in [100]$, $l = 3$, $p_1 = 0.4$, $p_2 = 0.35$, and $p_3 = 0.25$.

Thereby, $(ix)$ holds as all summands that are introduced by $\sum_{b'+r'=j_1-1} \binom{i-1}{b'} \binom{j_1-k}{r'}$ are zero. The only non-zero summand is for $b' = b$ and $r' = r$, that is, $b + r = j_1 - 1$. In general, $\binom{m}{n} = 0$ if $0 \leq m < n$ for $m \in \mathbb{R}$ and $n \in \mathbb{N}_0$.

$\square$

## C  External Theorems

This section collects identities and theorems that we use in the proofs of our statements.

**Definition C.1.** (Binomial coefficients; Graham et al. 1994, p. 154). When both arguments of a binomial coefficient are natural, one defines

$$\binom{n}{k} = \frac{n!}{k!\,(n-k)!},$$

for $n, k \in \mathbb{N}_0$ with $n \geq k \geq 0$. One extends this definition using falling factorials $r^{\underline{m}} := \prod_{i=0}^{m-1} (r-i)$ for any $r \in \mathbb{R}$ and $m \in \mathbb{N}_0$. Binomial coefficients are then expressed as

$$\binom{r}{m} = \frac{r^{\underline{m}}}{m!},$$

which allows for real arguments $r$.

**Theorem C.2.** (Binomial theorem; Graham et al. 1994, p. 162). *Given $x, y \in \mathbb{R}$ and $n \in \mathbb{N}_0$, the binomial theorem states*

$$(x + y)^n = \sum_{k=0}^{n} \binom{n}{k} x^{n-k} y^k.$$

**Theorem C.3.** (Graham et al. 1994, p. 164). *As a special case of the binomial theorem, one obtains*

$$(-1)^k \binom{-r}{k} = \binom{r + k - 1}{k},$$

*for $r \in \mathbb{R}$ and $k \in \mathbb{N}_0$.*

**Theorem C.4.** (Graham et al. 1994, p. 168). *Given $r \in \mathbb{R}$ and $m, k \in \mathbb{N}_0$ with $m \geq k$, it holds that*

$$\binom{r}{m}\binom{m}{k} = \binom{r}{k}\binom{r-k}{m-k}.$$

**Theorem C.5.** (Chu-Vandermonde identity; Graham et al. 1994, p. 169). *Given $s, t \in \mathbb{R}$ and $n \in \mathbb{N}_0$, the Chu-Vandermonde identity states*

$$\binom{s+t}{n} = \sum_{k=0}^{n} \binom{s}{k}\binom{t}{n-k}.$$

## D  Additional Experiments

This section contains additional experiments and details their hyperparameters. Section D.1 provides experiments justifying Assumption 4.4, Section D.2 lists the values of all relevant hyperparameters, the parameter sensitivity of our approach is discussed in Section D.3, and Section D.4 contains reject trade-off plots for all experimental settings considered.

### D.1  Applicability of Assumption 4.4

Given a fixed instance $\tilde{x}$, Assumption 4.4 describes the label distribution of the neighbors' candidate sets. Thereby, we assume that the true label $\tilde{y}$ dominates the neighborhood and that label $\tilde{y}_c$ co-occurs most often with the true label $\tilde{y}$. Otherwise, we assume uniform noise among the remaining noise labels.

Figure 3 demonstrates that those assumptions are largely satisfied on the four real-world datasets (see Section 5.3). In most cases, the true label $\tilde{y}$ occurs most often in the neighborhood of instance $\tilde{x}$. There are one or two other labels with which $\tilde{y}$ is commonly confused, which we model by $\tilde{y}_c$. Apart from that, all remaining noise labels are distributed uniformly.

### D.2  Hyperparameters

As mentioned in Section 5.1, we consider ten commonly used PLL approaches. We choose their parameters as recommended by the respective authors.

- PL-KNN (Hüllermeier & Beringer, 2005): For all non-*MNIST* datasets, we use $k = 10$ neighbors as recommended by the authors. For the *MNIST* datasets, we use the hidden representation of a variational auto-encoder as instance features and use $k = 20$. The variational auto-encoder has a 768-dimensional input layer (flat *MNIST* input), a 512-dimensional second layer, and 48-dimensional bottleneck layers for the mean and variance representations. The decoder uses a 48-dimensional first layer, a 512-dimensional second layer, and a 768-dimensional output layer with sigmoid activation. Otherwise, we use ReLU activations between all layers. Binary cross-entropy is used as a reconstruction loss. We choose the *AdamW* optimizer for training.

- PL-SVM (Nguyen & Caruana, 2008): We use the PEGASOS optimizer (Shalev-Shwartz et al., 2007) and $\lambda = 1$.

- IPAL (Zhang & Yu, 2015): We use $k = 10$ neighbors, $\alpha = 0.95$, and 100 iterations.

- PL-ECOC (Zhang et al., 2017): We use $L = \lceil 10 \log_2(l) \rceil$ and $\tau = 0.1$ as recommended.

- PRODEN (Lv et al., 2020): For a fair comparison, we use the same base models for all neural-network-based approaches. We use a standard $d$-300-300-300-$l$ MLP (Werbos, 1974) for the non-*MNIST* datasets with ReLU activations, batch normalizations, and softmax output. For the *MNIST* datasets, we use the LeNet-5 architecture (LeCun et al., 1998). We choose the *Adam* optimizer for training.

- CC (Feng et al., 2020): We use the same base models as mentioned above for PRODEN.

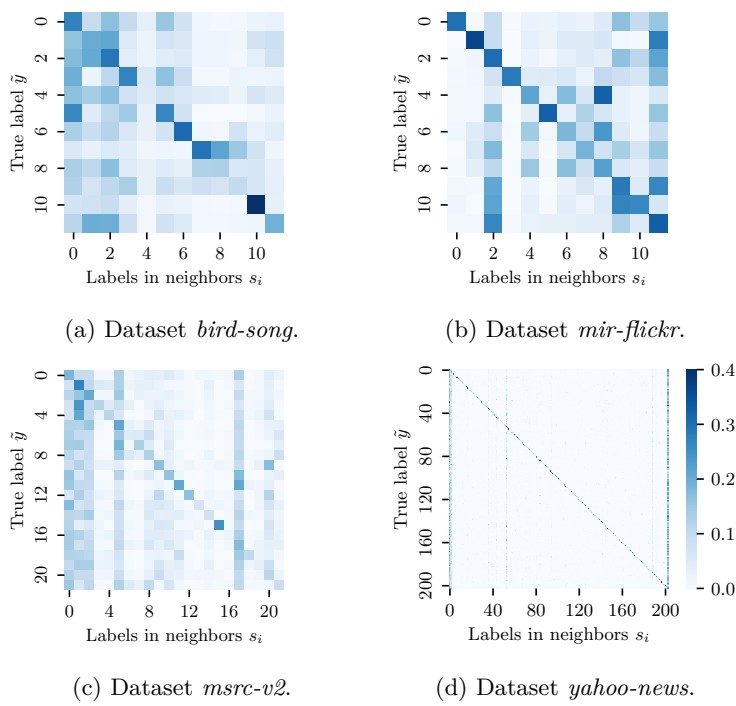

(a) Dataset *bird-song*.

(b) Dataset *mir-flickr*.

(c) Dataset *msrc-v2*.

(d) Dataset *yahoo-news*.

Figure 3: Counts of the class labels in the candidate sets $s_i$ of instance $\tilde{x}$'s 10-nearest neighbors $x_i$.

- VALEN (Xu et al., 2021): We use the same base models as mentioned above for PRODEN.

- POP (Xu et al., 2023): We use the same base models as mentioned above for PRODEN. Also, we set $e_0 = 0.001$, $e_{end} = 0.04$, and $e_s = 0.001$. We abstain from using the data augmentations discussed in the paper for a fair comparison.

- CROSEL (Tian et al., 2024): We use the same base models as mentioned above for PRODEN. We use 10 warm-up epochs using CC and $\lambda_{cr} = 2$. We abstain from using the data augmentations discussed in the paper for a fair comparison.

- DST-PLL (our proposed approach): Similar to PL-KNN and IPAL, we use $k = 10$ neighbors for the non-*MNIST* datasets. For the *MNIST* datasets, we use the hidden representation of a variational auto-encoder as instance features and use $k = 20$. The architecture of the variational auto-encoder is the same as described above for PL-KNN.

We have implemented all approaches in PYTHON using the PYTORCH library. All experiments need two to three days on a machine with 48 cores and one NVIDIA GeForce RTX 3090.

## D.3 Parameter Sensitivity

Figure 4 shows the sensitivity of the number of neighbors $k$ regarding the test-set performance, the fraction of confident / non-rejected predictions, and the non-rejected prediction performance. The shaded areas indicate the standard deviation regarding the 5-fold cross-validation. As for default $k$-nearest neighbor classification, changes of $k$ have a relatively large impact. We show parameter sensitivity for each of the real-world datasets separately. Naturally, different datasets have different optimal parameter settings. The configuration $k = 10$, which is also recommended within PL-KNN (Hüllermeier & Beringer, 2005) and IPAL (Zhang & Yu, 2015), provides a good trade-off between the number of confident predictions and how accurate confident predictions are. Indeed, this setting produces a good number of confident predictions on most datasets (Figure 4; center plot). At the same time, it produces a good MCC performance of confident predictions on most datasets (Figure 4; right plot).

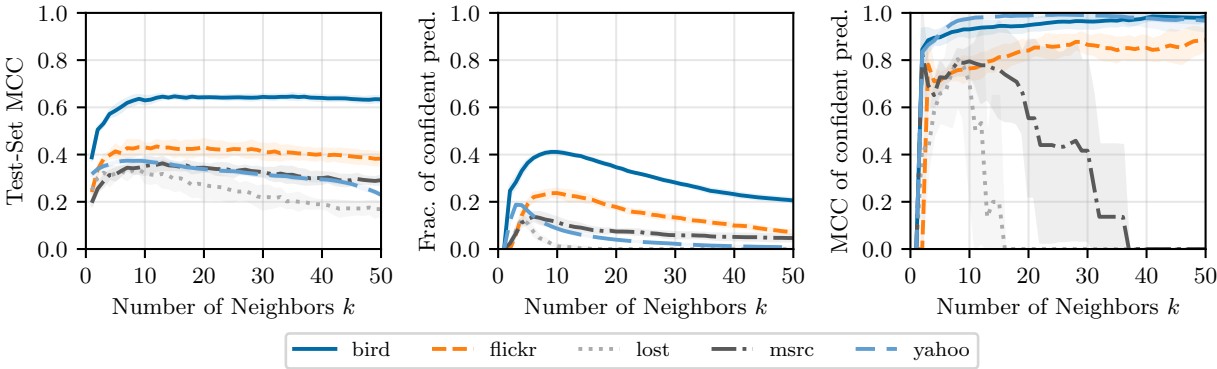

Figure 4: Sensitivity of $k$ regarding the test-set MCC score, the fraction, and the MCC score of confident / non-rejected predictions.

When increasing $k$, our method's behavior on different datasets can be assembled into two groups. On the *bird-song*, *mir-flickr*, and *yahoo-news* datasets, increasing $k$ past ten neighbors also increases the amount of irrelevant labeling information from those neighbors. Therefore, our approach produces less confident predictions. At the same time, the MCC score of confident predictions remains at roughly the same level. This is because irrelevant labeling information from neighbors increases at most at the same rate as $k$. In contrast, on the *lost* and *msrc-v2* datasets, the MCC score of confident predictions drops sharply at a certain point while the number of confident predictions decreases similarly. This is because irrelevant labeling information increases more rapidly than $k$: The decrease of confidence in predictions is slower than the increase of irrelevant candidate labels.

### D.4   Reject Trade-off Curves

Figure 5 (i) – (xxxiv) shows the reject trade-off for varying confidence (0 to 1) and $\Delta_{\tilde{m}}$ (-1 to 1) thresholds and augments Figure 1 by considering all datasets and noise generation strategies. The x-axes show the fractions of predictions that are rejected. The y-axes show the accuracies of predictions that are not rejected. The plots show (fraction of rejects, non-rejected test-set accuracy)-pairs corresponding to different settings of the thresholds. In most cases, our method provides a better trade-off between the number of rejected predictions and the accuracy of the non-rejected predictions. Table 2 summarizes all plots by showing the average empirical risks across all experimental settings and for different trade-off parameters $\lambda$. We recall that our method provides the significantly best trade-offs for $\lambda \in [0, 0.2]$.

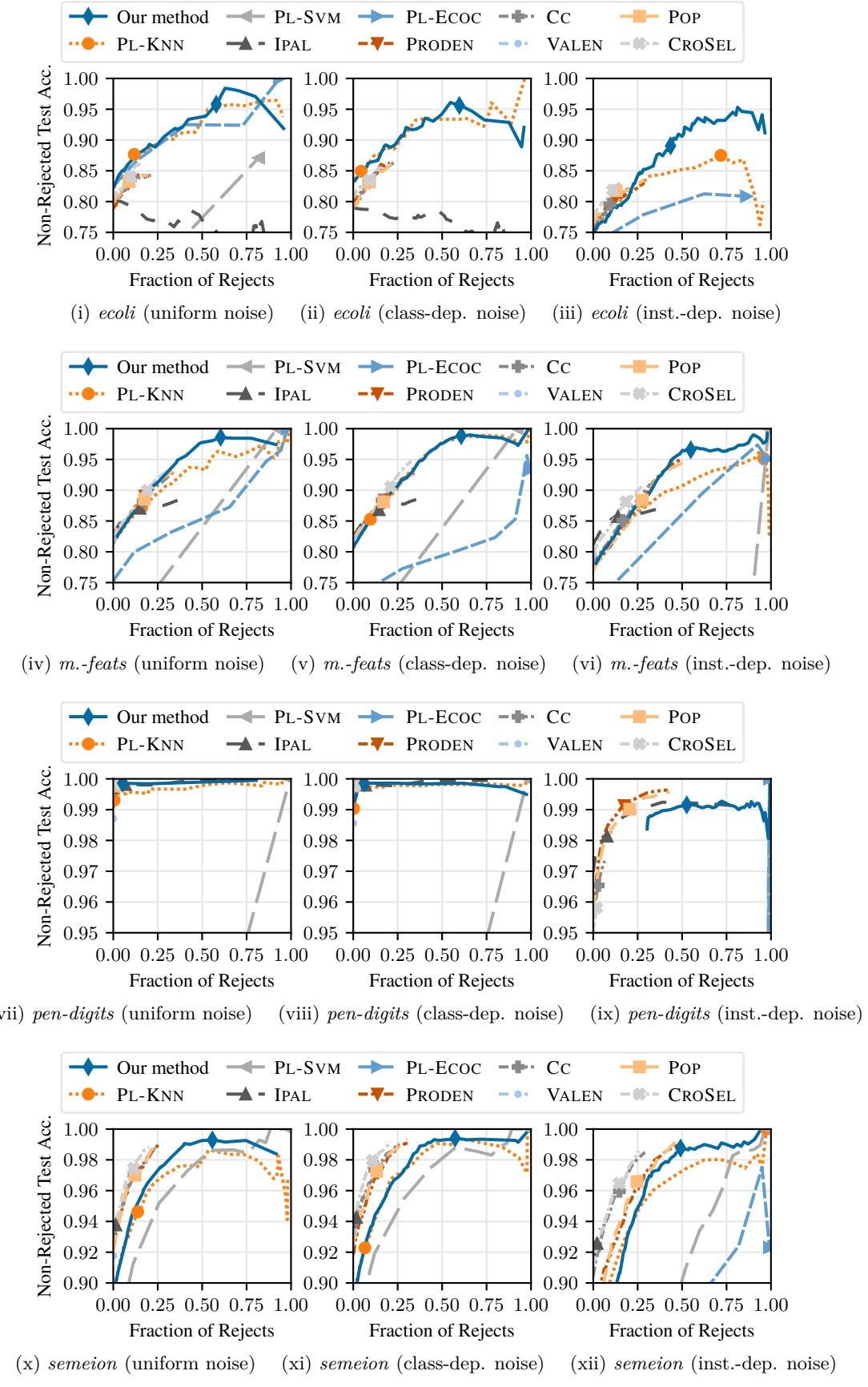

(i) *ecoli* (uniform noise)    (ii) *ecoli* (class-dep. noise)    (iii) *ecoli* (inst.-dep. noise)

(iv) *m.-feats* (uniform noise)    (v) *m.-feats* (class-dep. noise)    (vi) *m.-feats* (inst.-dep. noise)

(vii) *pen-digits* (uniform noise)    (viii) *pen-digits* (class-dep. noise)    (ix) *pen-digits* (inst.-dep. noise)

(x) *semeion* (uniform noise)    (xi) *semeion* (class-dep. noise)    (xii) *semeion* (inst.-dep. noise)

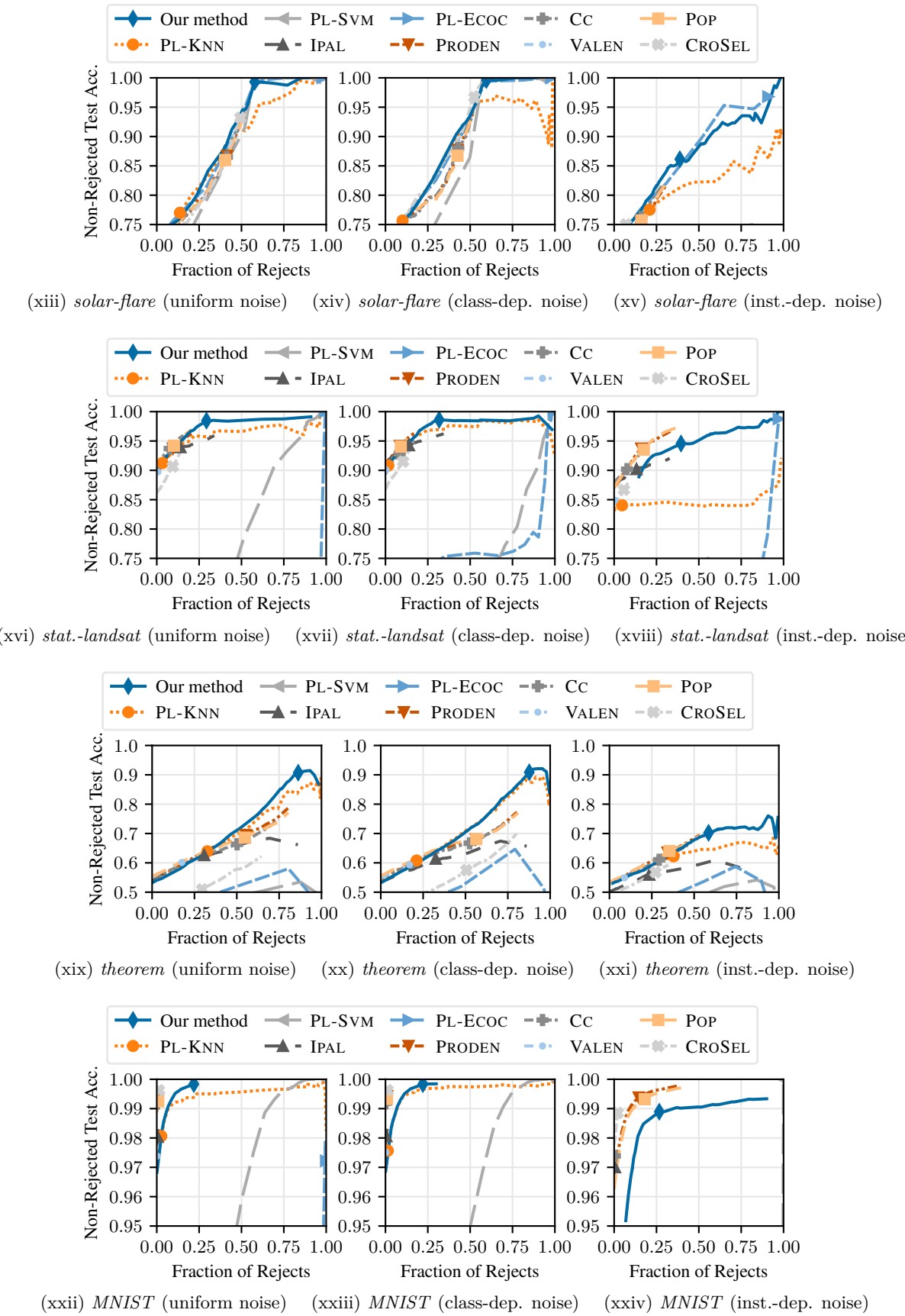

(xiii) *solar-flare* (uniform noise)  (xiv) *solar-flare* (class-dep. noise)  (xv) *solar-flare* (inst.-dep. noise)

(xvi) *stat.-landsat* (uniform noise)  (xvii) *stat.-landsat* (class-dep. noise)  (xviii) *stat.-landsat* (inst.-dep. noise)

(xix) *theorem* (uniform noise)  (xx) *theorem* (class-dep. noise)  (xxi) *theorem* (inst.-dep. noise)

(xxii) *MNIST* (uniform noise)  (xxiii) *MNIST* (class-dep. noise)  (xxiv) *MNIST* (inst.-dep. noise)

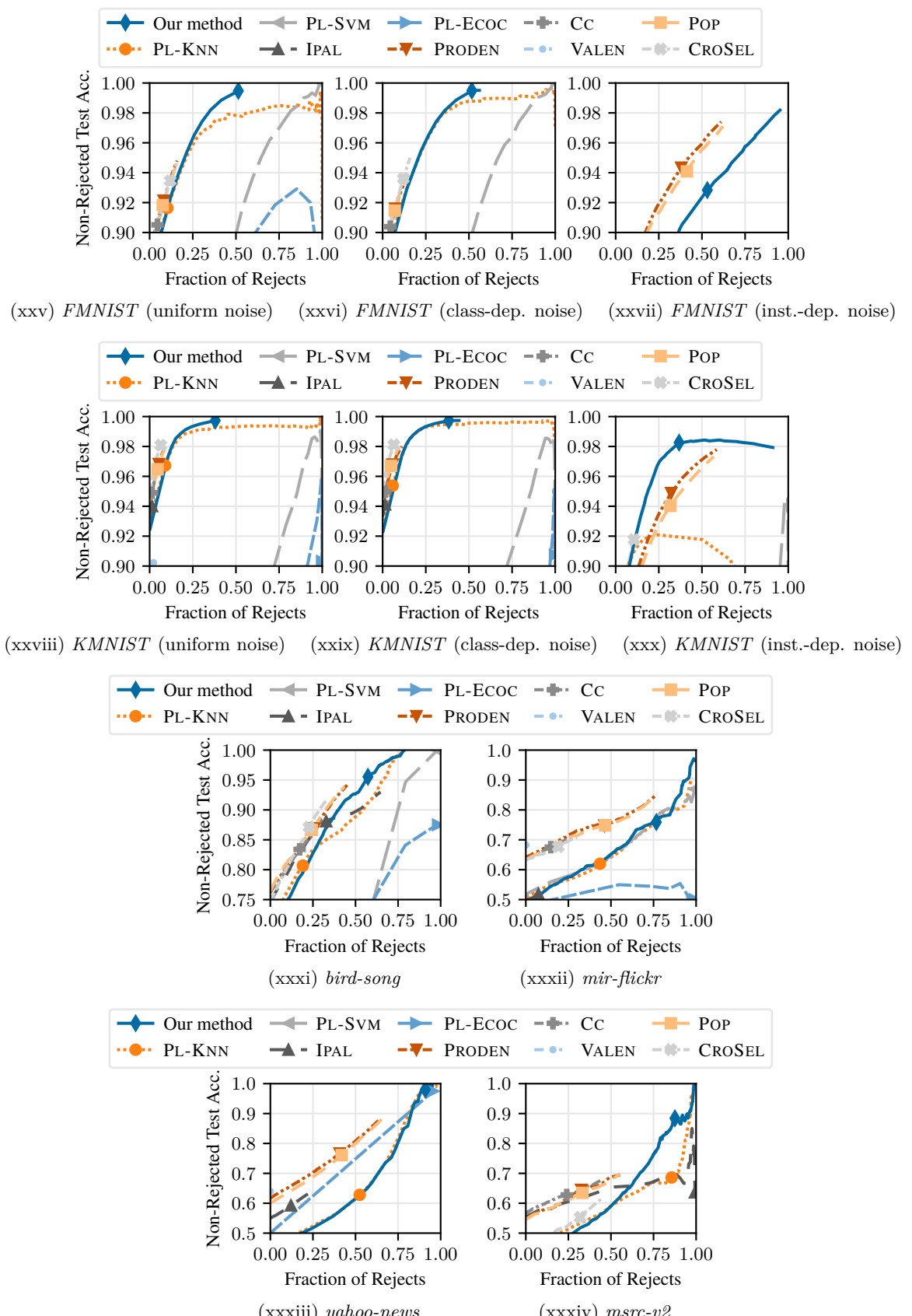

Figure 5: Trade-off between the fraction of rejected predictions and the accuracy of non-rejected predictions.

