# OpenReview forum: "Partial-Label Learning with a Reject Option"
_TMLR — Accepted by TMLR_

### Review · Reviewer_xj75 · 2024-10-25

**Summary Of Contributions:**

The paper deals with learning a classifier with a reject option under partial labels. The authors propose to use a KNN classifier, where the class probabilities are adjusted based on Dempster-Shafer Theory (DST) and combined using Yager's rule. The proposed confidence-based rejection strategy encodes that high probability mass is required for a class prediction if other viable options exist, but lower probability mass is sufficient if there are no other significantly competitive options.

The approach is evaluated against 10 SOTA partial label learning approaches on supervised datasets augmented with noise (including datasets from UCI and MNIST variants) as well as partial label datasets (including bird-song, flickr. yahoo-news). The authors evaluate approach without reject option as well with it. The results show that the approach is on par with SOTA approaches if no reject option is used, but improves the results when reject options are used.

**Audience:**

Yes

**Broader Impact Concerns:**

There are no broader impact concerns.

**Claims And Evidence:**

Yes

**Requested Changes:**

Please clarify the following points:
* How would the reject option for other the other approaches perform if confidences where calibrated or if the best threshold would be searched for? It is partially visible in Figure 1, but not for all results in the table.
* How can I interpret the results on the far right in Figure 1? Does this end with > 99% but < 100% of labels rejected?
* Couldn't one use the reject option approach also for the other approaches and if yes, would it increase their performance?
* How does the approach, in general, relate to calibration of classifiers? It would refer to the related work section on the confidence-based approach, but only bayesian methods and ensembles are mentioned.

Lastly, it would be beneficial to have a designated "Conclusions" section to summarize the results instead of ending the paper with the section on results.

**Strengths And Weaknesses:**

# Strengths
* The tackled problem is important fits very well to the venue.
* The paper presentation is sound - it is sufficiently format, includes analyses such as for the runtime and consistency.
* The evaluation setup is sufficiently broad in terms of used comeptitors and datasets

# Weaknesses
* Unclear if the proposed method is adding value based on the empirical results. The base approach is only on par with other approaches without reject option
* Unclear how the the proposed reject option relates to calibration methods and how it would perform against calibrated classifiers

---

> ### Author Response · Authors · 2024-10-28
>
> We thank the reviewer for the time and effort invested. Below, we answer your comments and questions in detail.
>
> **Added value based on the empirical results.**
> We agree that our base approach (without reject option) matches the predictive performance of the competitors.
> Still, we emphasize that our approach with reject option achieves significantly better empirical risks (Table 2) for various trade-off values of $\lambda$.
> As is evident from these results, our method is especially beneficial if misclassifications are costly (small values of $\lambda$), which is the setting that we target (Section 1).
>
> **Relation to calibration methods.**
> We note that while reject options provide a binary decision, calibration methods modify the predicted confidences such that they align with the observed accuracies.
> In this sense, both approaches are orthogonal and cannot directly be compared.
> Formally, we adapt $\theta$ to achieve desirable properties (Theorem 4.1) of the resulting reject option $\Delta = conf(g) - \theta$.
> In contrast to that, calibration methods adapt the confidences $conf(g)$; to obtain a corresponding reject option, one still needs to fix $\theta$.
> In our experiments, we fix a sensible threshold for the competitors (indicated by the markers in Figure 1); this step is not necessary when using our proposed method as $\theta$ is chosen adaptively based on the data.
> We will emphasize this point in the updated version of the manuscript.
>
> **Interpretation of Figure 1.**
> Recall from the caption of Figure 1 that the plots show (fraction of rejects, non-rejected test set accuracy)-pairs corresponding to different settings of the thresholds.
> The top-right corner of the plots generally describes the case when almost all predictions are rejected and the few accepted predictions are mostly correct.
> E.g., in Figure 1(c), our approach reaches perfect test-set accuracy when rejecting about 99% of all predictions for the dataset shown; the point (1, 1) cannot be observed as the test-set accuracy is undefined if all predictions are rejected.
>
> **Other methods with our reject option.**
> Our confidence-based reject option relies on Dempster-Shafer theory to adaptively choose the reject threshold $\theta$ based on the evidence extracted from the nearest neighbors.
> While it is, in general, possible to use confidence-based reject options with any of the used methods, our concrete strategy depends on the underlying k-NN classifier and the combination of the neighbors' evidence.
> It is not straightforward how to extend our reject option to the other cases.
>
> **Missing conclusions section.**
> Thank you for your comment.
> We will add a conclusions section to our paper and update the uploaded manuscript once all three reviews are submitted.
>
> We hope this rebuttal answers your questions.

---

> ### Author Response · Authors · 2024-11-12
> **Uploaded Revision**
>
> We again thank the reviewer for their time invested.
> We are happy to announce that we have uploaded a new version of the manuscript.
> As suggested, we made the following changes.
>
> * We have added a discussion about calibration methods in the related work section (Section 2.2) to highlight similarities and differences to reject options.
> * We have improved the discussion of Figure 1, simplifying its interpretation.
> * We have added new experimental results in Figure 5 (Appendix D.4). These augment the results in Figure 1 by considering all experimental settings. With this more diverse range of settings, the take-away stays the same: Our method yields the best trade-offs for many experimental settings and gives the best average trade-off risks (Table 2).
> * We have added a new experiment (Table 3) with fixed reject rates: When tuning the competitors' thresholds to reject a similar number of instances as our method, our algorithm achieves the best average accuracies across all experimental settings.
> * We have added a Conclusions section (Section 6).
>
> We hope that these additions and changes settle all open questions.

---

### Review · Reviewer_z1xF · 2024-10-29

**Summary Of Contributions:**

This paper focus on the limitation of potentially incorrect prediction of current partial-label learning methods. The authors utilize a nearest-neighbor algorithm to avoid the risk of incorrect prediction and propose a confidence-based strategy. They also conduct a series of experiment to validate the effectiveness of proposed method.

**Audience:**

Yes

**Claims And Evidence:**

Yes

**Requested Changes:**

Please find in weaknesses.

**Strengths And Weaknesses:**

Pros:
1. This paper is easy to follow.
2. Authors provide theoretical justification and analysis for the proposed method.

Cons:
1. The authors assumpt that the near samples in feature space attribute to the reliability of the target sample, and thus a reject option. But it is more from an intuition and lacks of theoretical justification or experimental results.
2. The novelty. The reject option machinical has been studied in recent years. The authors seems to simply apply it on the PLL tasks.
3. The experimental results in Table 1 could not validate the effectiveness of proposed method.

---

> ### Author Response · Authors · 2024-10-31
>
> We thank the reviewer for the time and effort invested. Below, we answer your comments and questions in detail.
>
> **Neighborhood assumption.**
> Indeed, Assumption 4.2 relies on an intuitive assumption about the neighborhood of an instance.
> Assumptions of this form are common in the literature when analyzing PLL algorithms.
> For example (see also Section 4.3):
>
> * Cour et al. (2011) make similar assumptions (Definition 2: Ambiguous distribution; Proposition 5: Dominance relation), which bound the noise from incorrect candidate labels.
> * Feng et al. (2020) make a stronger assumption by assuming a uniform distribution of the incorrect candidate labels.
> * More assumptions of this type can be found in Liu & Dietterich (2012) and Lv et al. (2020).
>
> We remark that some assumption about the candidate label distribution (like Assumption 4.2) is necessary to give statistical guarantees in the PLL setting (Lemma 4.3 and Theorem 4.4).
> We are happy to include this extended discussion in the revised manuscript.
>
> **Novelty.**
> Our confidence-based reject option relies on Dempster-Shafer theory to adaptively choose the reject threshold $\theta$ based on the evidence collected from the nearest neighbors.
> To the best of our knowledge, our manuscript is the first to propose such a reject option in the PLL setting.
> While partially-labeled data is especially challenging to handle (incorrect candidate labels can easily mislead the classifier), we prove that our novel reject option satisfies various desirable properties (Theorem 4.1).
> Hence, we believe that our article makes algorithmic and theoretical contributions.
>
> **Experimental results.**
> Table 1 shows the predictive performance of the base classifier without our proposed reject option.
> Our base approach (without reject option) matches the predictive performance of the competitors.
> Still, we emphasize that our approach with reject option achieves significantly better empirical risks (Table 2) for various trade-off values of $\lambda$; compare Equation (1).
>
> We hope this rebuttal answers all your questions.

---

> ### Author Response · Authors · 2024-11-12
> **Uploaded Revision**
>
> We again thank the reviewer for their time invested.
> We are happy to announce that we have uploaded a new version of the manuscript.
>
> * As suggested, we have improved the discussion of Assumption 4.4 by highlighting similar assumptions made in the literature and by explicitly pointing out its necessity.
> * We have included new experimental results in Figure 5 (Appendix D.4), which contains similar plots to Figure 1 but considers all our experimental settings: Our method yields the best trade-offs for many experimental settings and gives the best average trade-off risks (Table 2).
> * We have added a new experiment (Table 3) with fixed reject rates: When tuning the competitors' reject thresholds to reject a similar number of instances as our approach does, the proposed method achieves the best average accuracies across all experimental settings.
>
> We hope these adjustments address all open questions.

---

### Review · Reviewer_xnXJ · 2024-11-05

**Summary Of Contributions:**

The authors consider the problem of Partial Label Learning (PLL), where the training samples contain instances with multiple candidate labels, of which one is the true label. They propose a new nearest neighbor (kNN) style learning algorithm that allows the classifier to optionally *abstain* from making a prediction.

Having found the $k$-nearest neighboring instances to the current test instance and their associated candidate label sets, the algorithm uses Dempster-Shafer theory (DFT) to predict a label and its associated confidence score. By thresholding the confidence score, they then decide whether to return the predicted label or abstain from making a prediction.

DFT is an alternative to classical probability theory for reasoning about uncertain outcomes. In the context of this paper, each of the $k$ candidate label sets is seen as a source of evidence, and are associated with a mass function that assigns probabilities to subset of labels. Applying tools from DFT, the authors aggregate the $k$ mass functions to derive both a predicted label and a confidence score.

**Audience:**

Yes

**Claims And Evidence:**

Yes

**Requested Changes:**

I would be more inclined to accepting the paper if the authors address the following questions/comments:

- **Justification for probability assignments in Algorithm 1:**

The choice of basic probability assignments $m_i$s needs further justification. Taking $\tilde{s} = \mathcal{Y}$ for an unseen test instance, your choosen $m_i$ assigns a probability of 1/2 to both the set $s_i$ and the set of all labels $ \mathcal{Y}$. The paper mentions why you choose equal probabilities for the two sets, but whats unclear to me is why $m_i$ assigns non-zero probabilities to only two sets. I could imagine a probability assignment where  $s_i$ receives a probability of 1/2 and all other possible subsets other than $s_i$ receive equal probabilities, such that $m_i$ totals to 1.

Could the authors elaborate on their rationale behind $m_i$ having a support of size 2?

Furthermore, wouldn't it be more accessible to present Algorithm 1 for an unseen test instances $\tilde{x}$, instead of a generic $\tilde{s}$, thereby simplifying lines 3-7 to only the Else part.

- **Illustrations to convey intuition:**

I would urge the authors to include a simple toy example to go over the workings of their algorithm. For example, a particular corner case of interest is when the candidate label sets have exactly one label in common - it would be useful to show that your method would indeed predict this label with high confidence.

The following is an example that helped me understand and work through your method. Suppose the candidate label sets from k=3 neighbors are {1, 2}, {1, 3}, {1, 4}, then the belief probability assignment $\tilde{m}$ from (4) evaluates to:

{1}:  1/2

{1, 2}:  1/8

{1, 3}:  1/8

{1, 4}:  1/8

{1, 2, 3, 4}:  1/8

Its intuitive that because 1 is the lone common label in across candidate sets, it receives a higher probability mass. The classification rule predicts label 1. The rejection confidence score evaluates to 1/2 - 1/4 = 1/4 > 0; therefore, we accept the prediction.

- **Intuitive background description for DFT:**

Please elaborate on Sec 3.2 to provide a more intuitive and self-contained discussion on DFT.

- **Experimental metrics:**

The experimental results in Table 1 and Table 2 use particular thresholds for different methods (50% for some baselines, 90% for proposed method). Instead, for a more fair comparison, I would strongly recommend tuning the thresholds to reach a fixed rejection rate and reporting test accuracies for all methods at that fixed rejection rate. In other words, reporting accuracies for different fixed rejection rates may be a more useful comparison than fixing the thresholds and varying the rejection cost $\lambda$.

**Strengths And Weaknesses:**

Strengths:
- The paper addresses an important problem and appears to be the first to introduce a "reject option" under the PLL setting.
- The use of DFT to aggregate candidate label sets into a predicted label and confidence score is novel and very interesting.
- Experiments suggest that the proposed approach provide better trade-offs between accuracy and rejection rates compared to the baselines.
- The authors provide theoretical consistency results for their method under some specific distributional assumptions.

Weaknesses:
- Some of the design choices in applying DFT to PLL have not been aptly justified. As a reader new to DFT, many elements of the proposed algorithm seemed initially to be arbitrarily chosen. Only upon reading background material on DFT, was I able to convince myself of the rationale behind the design choices made (see changes requested).

- The paper fails to provide sufficient intuition for why the confidence measure or the label prediction rule proposed is suitable for PLL. Yes, the proposed algorithm is derived using principles from DFT, but the writing could provide some concrete intuition for why one would expect the method to work in practice (see changes requested).

- Finally, the paper could do a better job of providing background on DFT. Currently, Section 3.2 is short and packed with notations and jargons, making it hard for an unfamiliar reader to grasp the basics of DFT.

---

> ### Author Response · Authors · 2024-11-06
>
> We thank the reviewer for the time and effort invested and the detailed review. Below, we answer your comments and questions in detail.
>
> **Justification of probability assignments in Algorithm 1.**
> Recall that basic probability assignments $m$ in Dempster-Shafer theory (DST) assign probability mass to subsets of the label space $\mathcal{Y}$ and satisfy $\sum_{A \in 2^{\mathcal{Y}}} m(A) = 1$.
> This assignment differs from standard probability as $m(\mathcal{Y}) \leq 1$ and independent events do, generally, not sum up, e.g., $m(\lbrace1, 2\rbrace) \neq m(\lbrace1\rbrace) + m(\lbrace2\rbrace)$.
> DST allows for more flexibility as one can allocate mass on $\lbrace1, 2\rbrace$ without needing to specify the mass of $\lbrace1\rbrace$ and $\lbrace2\rbrace$ if uncertain.
> For example, let $\mathcal{Y} = \lbrace1, 2, 3\rbrace$ and $s_i = \lbrace1, 2\rbrace$.
> As you noted, our method constructs $m_i(\lbrace1, 2, 3\rbrace) = 1/2$ and $m_i(\lbrace1, 2\rbrace) = 1/2$; otherwise $m_i(A) = 0$.
>
> We compare our proposed method and your suggested strategy below:
>
> *Our strategy.* When labeling a previously unseen instance $x$, we look at its nearest neighbors $(x_i, s_i)$.
> The case $m_i(\lbrace1, 2\rbrace) = 1/2$ represents that the correct label of $x$ is likely hidden within its neighbor's candidates.
> The case $m_i(\lbrace1, 2, 3\rbrace) = 1/2$ represents that the neighbor $x_i$ does not explain the labeling of $x$: we allocate probability mass on the full label space to reflect full uncertainty.
> For reference, note that, in the supervised case, Denoeux (1995) use $m_i(s_i) = \alpha$, $m_i(\mathcal{Y}) = 1 - \alpha$, and $m_i(A) = 0$ otherwise.
> However, we target a more general setting: Our focal sets can be arbitrary subsets instead of only singletons or the full label set.
>
> *Suggested strategy.* If one, instead, defined $m_i(\lbrace1, 2\rbrace) = 1/2$ and $m_i(\lbrace1\rbrace) = m_i(\lbrace2\rbrace) = m_i(\lbrace3\rbrace) = m_i(\lbrace1, 3\rbrace) = m_i(\lbrace2, 3\rbrace) = m_i(\lbrace1, 2, 3\rbrace) = 1/12$, one would allocate probability mass to incorrect labels, that is, $\lbrace1\rbrace$ does not include the correct label if the correct label is $2$, $\lbrace2\rbrace$ does not include the correct label if the correct label is $1$, $\lbrace3\rbrace$ does never include the correct label, $\lbrace1, 3\rbrace$ does not include the correct label if the correct label is $2$, and $\lbrace2, 3\rbrace$ does not include the correct label if the correct label is $1$.
> In contrast to that, in our approach, $\lbrace1, 2\rbrace$ and $\lbrace1, 2, 3\rbrace$ always contain the correct label $y_i \in s_i$: $m_i$ only allocates mass on events that we have evidence on, i.e., that we surely know about.
>
> Using Yager's combination rule, we then combine and refine all basic probability assignments $m_i$ into $\tilde{m}$ (Section 4.1).
> The combined basic probability assignment $\tilde{m}$ has generally more than two subsets with non-zero mass.
>
> We will include this extended discussion and an example in the revised manuscript.
>
> **Intuition of DST.**
> The given example captures the application of our method well.
> We are happy to include an extended discussion and an example to build intuition in the revised manuscript.
> Also, we will emphasize the difference to standard probability and highlight the strengths of DST.
>
> **Experimental metrics.**
> We have conducted additional experiments with a fixed rejection rate and will include the results in the revised manuscript.
> To obtain the results, we tuned the reject thresholds such that all methods achieve a similar number of rejects.
> Still, our method has superior test accuracy on the non-rejected predictions.
>
> | Algorithms     | Fraction of rejects | Non-rejected test accuracy |
> |----------------|---------------------|----------------------------|
> | Pl-Knn (2005)  | 0.502 ($\pm$ 0.210) | 0.912 ($\pm$ 0.101)        |
> | Pl-Svm (2008)  | 0.502 ($\pm$ 0.210) | 0.744 ($\pm$ 0.198)        |
> | Ipal (2015)    | 0.502 ($\pm$ 0.210) | 0.835 ($\pm$ 0.161)        |
> | Pl-Ecoc (2017) | 0.502 ($\pm$ 0.210) | 0.739 ($\pm$ 0.171)        |
> | Proden (2020)  | 0.502 ($\pm$ 0.210) | 0.940 ($\pm$ 0.083)        |
> | Cc (2020)      | 0.502 ($\pm$ 0.210) | 0.901 ($\pm$ 0.179)        |
> | Valen (2021)   | 0.502 ($\pm$ 0.210) | 0.868 ($\pm$ 0.130)        |
> | Pop (2023)     | 0.502 ($\pm$ 0.210) | 0.940 ($\pm$ 0.082)        |
> | CroSel (2024)  | 0.502 ($\pm$ 0.210) | 0.897 ($\pm$ 0.187)        |
> | Dst-Pll (ours) | 0.501 ($\pm$ 0.211) | **0.955** ($\pm$ 0.070)    |
>
> We hope this rebuttal answers all your questions.
> We will upload the revised manuscript soon.

---

> ### Author Response · Authors · 2024-11-12
> **Uploaded Revision**
>
> We again thank the reviewer for their time invested.
> We are happy to announce that we have uploaded a new version of the manuscript.
>
> * As suggested by the reviewer, we have improved the background section on DST (Section 3.2) to ease its understanding.
> * We have extended the discussion of our approach in the main section (Section 4.1 and 4.2) by adding more explanation and two novel examples (Example 1 and Example 2) for the classification and reject rules.
> * We have added a new experiment (Table 3) with fixed reject rates: When tuning the competitors' thresholds to reject a similar number of instances as our method, the proposed algorithm achieves the best average accuracies across all experimental settings.
> * We have included new experimental results in Figure 5 (Appendix D.4), which contains plots similar to those in Figure 1 but for all our experimental settings. In this larger-scale experiment, our method still obtains the best trade-off across many experimental settings and gives the best average trade-off risks (Table 2).
>
> We hope our changes, particularly the extended experiments, settle all open questions.

---

> > ### Comment · Reviewer_xnXJ · 2024-12-13
> > **Re: Uploaded Revision**
> >
> > I appreciate the authors' clarifications and updates to the paper.
> >
> > A minor typo: "Also, mass allocated to non-intersecting sets" -> "Also, **the** mass allocated to non-intersecting sets".
> >
> > It would also be good to give more intuitive descriptions early on for the terms introduced in Sec 3.2: for example, "belief" and "plausibility" are introduced as equations in (2) without clearly explaining what these concepts mean in DFT (you do get to this a bit later). Also, it might be good to state early on that DFT allows us to combine evidences from multiple sources, and why this setup is a good fit for PLL.

---

> > > ### Author Response · Authors · 2024-12-17
> > >
> > > We thank the reviewer for their suggestions and addressed them in the updated version of the paper.
> > > More specifically, we have fixed the typo and extended the discussion of Section 3.2, providing a better intuition of the DST concepts.

---

### Decision · Action_Editor_KMrx · 2024-12-23

**Recommendation:** Accept as is

**Comment:**

We uphold the positive reviewer consensus and recommend the paper for publication. The revised version has improved the manuscript in readability and scope of results, and is expected to be of interest to the community.

**Audience:**

Partial-label learning is a problem of fundamental interest in the weakly supervised learning literature. The present paper offers a new way of framing the problem, which is demonstrated to have theoretical and practical benefits. This is expected to be of broad interest to those working in the area.

More generally, the paper could be of interest to researchers interested in practical applications of DST in machine learning.

**Claims And Evidence:**

The paper's central claim is that one can effectively address the problem of partial-label learning with a reject option with an algorithm based on Dempster-Shafer theory (DST). The basic idea is to use the partial label information to construct certain probability assignments in a DST setup. These are used to construct an adaptive confidence-based rejection rule, which has some provable guarantees, and has promising empirical performance.

Reviewers generally viewed the paper's claims favourably, with two exceptions:
- there were concerns about the DST framework not being described sufficiently clearly for those unfamiliar with it.
- there were concerns about the experimental results not showing gains over existing approaches.

For the first point, following the author response, the authors uploaded a revision that expanded Sec 3 with more background on Sec 3, as well as a concrete example to illustrate the intuition of the proposed algorithm. With this, the work is expected to be more accessible to readers.

For the second point, following the author response, the authors clarified that the proposed method yields most significant gains in settings where misclassifications are costly. The authors also included additional experiments with a fixed rejection rate, showing that the proposed method yields better accuracies on non-rejected samples. Overall, these are convincing on the practical utility of the proposed method.